# SELF-REFINE:
# Iterative Refinement with Self-Feedback

**Aman Madaan[1], Niket Tandon[2], Prakhar Gupta[1], Skyler Hallinan[3], Luyu Gao[1],
Sarah Wiegreffe[2], Uri Alon[1][*], Nouha Dziri[2], Shrimai Prabhumoye[4], Yiming Yang[1],
Shashank Gupta[2], Bodhisattwa Prasad Majumder[5], Katherine Hermann[6],
Sean Welleck[2,3], Amir Yazdanbakhsh[6], Peter Clark[2]**
[1]Language Technologies Institute, Carnegie Mellon University
[2]Allen Institute for Artificial Intelligence
[3]University of Washington   [4]NVIDIA   [5]UC San Diego [6]Google Deepmind
selfrefine@googlegroups.com

## Abstract

Like humans, large language models (LLMs) do not always generate the best output on their first try. Motivated by how humans refine their written text, we introduce SELF-REFINE, an approach for improving initial outputs from LLMs through iterative feedback and refinement. The main idea is to generate an initial output using an LLM; then, the same LLM provides *feedback* for its output and uses it to *refine* itself, iteratively. SELF-REFINE does not require any supervised training data, additional training, or reinforcement learning, and instead uses a single LLM as the generator, refiner, and feedback provider. We evaluate SELF-REFINE across 7 diverse tasks, ranging from dialog response generation to mathematical reasoning, using state-of-the-art (GPT-3.5 and GPT-4) LLMs. Across all evaluated tasks, outputs generated with SELF-REFINE are preferred by humans and automatic metrics over those generated with the same LLM using conventional one-step generation, improving by ∼20% absolute on average in task performance. Our work demonstrates that even state-of-the-art LLMs like GPT-4 can be further improved at test-time using our simple, standalone approach.[2].

## 1   Introduction

Although large language models (LLMs) can generate coherent outputs, they often fall short in addressing intricate requirements. This mostly includes tasks with multifaceted objectives, such as dialogue response generation, or tasks with hard-to-define goals, such as enhancing program readability. In these scenarios, modern LLMs may produce an intelligible initial output, yet may benefit from further iterative refinement—i.e., iteratively mapping a candidate output to an improved one—to ensure that the desired quality is achieved. Iterative refinement typically involves training a refinement model that relies on domain-specific data (e.g., Reid and Neubig (2022); Schick et al. (2022a); Welleck et al. (2022)). Other approaches that rely on external supervision or reward models require large training sets or expensive human annotations (Madaan et al., 2021; Ouyang et al., 2022), which may not always be feasible to obtain. These limitations underscore the need for an effective refinement approach that can be applied to various tasks without requiring extensive supervision.

Iterative *self*-refinement is a fundamental characteristic of human problem-solving (Simon, 1962; Flower and Hayes, 1981; Amabile, 1983). Iterative self-refinement is a process that involves creating an initial draft and subsequently refining it based on self-provided feedback. For example, when

---

[*]Now at Google DeepMind
[2]Code and data at https://selfrefine.info/

37th Conference on Neural Information Processing Systems (NeurIPS 2023).

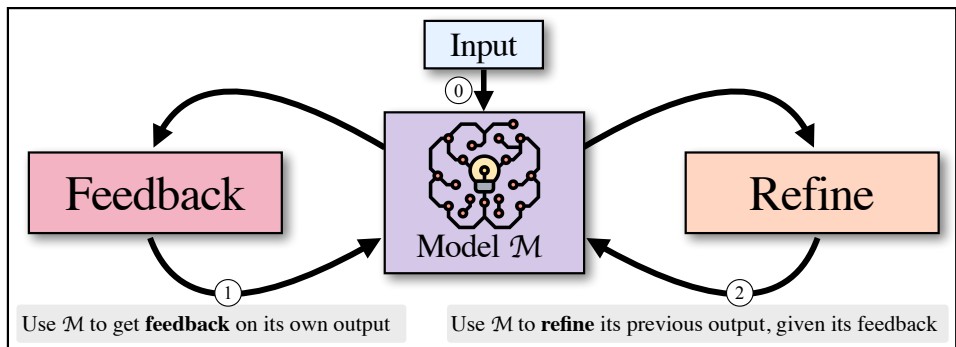

Figure 1: Given an input (⓪), SELF-REFINE starts by generating an output and passing it back to the same model $\mathcal{M}$ to get feedback (①). The feedback is passed back to $\mathcal{M}$, which refines the previously generated output (②). Steps (①) and (②) iterate until a stopping condition is met. SELF-REFINE is instantiated with a language model such as GPT-3.5 and does not involve human assistance.

drafting an email to request a document from a colleague, an individual may initially write a direct request such as "*Send me the data ASAP*". Upon reflection, however, the writer recognizes the potential impoliteness of the phrasing and revises it to "*Hi Ashley, could you please send me the data at your earliest convenience?*". When writing code, a programmer may implement an initial "quick and dirty" implementation, and then, upon reflection, refactor their code to a solution that is more efficient and readable. In this paper, we demonstrate that LLMs can provide iterative self-refinement without additional training, leading to higher-quality outputs on a wide range of tasks.

We present SELF-REFINE: an iterative self-refinement algorithm that alternates between two generative steps–FEEDBACK and REFINE. These steps work in tandem to generate high-quality outputs. Given an initial output generated by a model $\mathcal{M}$, we pass it back to the same model $\mathcal{M}$ to get *feedback*. Then, the feedback is passed back to the same model to *refine* the previously-generated draft. This process is repeated either for a specified number of iterations or until $\mathcal{M}$ determines that no further refinement is necessary. We use few-shot prompting (Brown et al., 2020) to guide $\mathcal{M}$ to both generate feedback and incorporate the feedback into an improved draft. Figure 1 illustrates the high-level idea, that SELF-REFINE *uses the same underlying language model to generate feedback and refine its outputs*.

We evaluate SELF-REFINE on 7 generation tasks that span diverse domains, including natural language and source-code generation. We show that SELF-REFINE outperforms direct generation from strong LLMs like GPT-3.5 (text-davinci-003 and gpt-3.5-turbo; OpenAI; Ouyang et al., 2022) and GPT-4 (OpenAI, 2023) by 5-40% absolute improvement. In code-generation tasks, SELF-REFINE improves the initial generation by up to absolute 13% when applied to strong code models such as CODEX (code-davinci-002; Chen et al., 2021). We release all of our code, which is easily extensible to other LLMs. In essence, our results show that even when an LLM cannot generate an optimal output on its first try, the LLM can often provide useful feedback and improve its own output accordingly. In turn, SELF-REFINE provides an effective way to obtain better outputs from a single model without any additional training, via iterative (self-)feedback and refinement.

## 2 Iterative Refinement with SELF-REFINE

Given an input sequence, SELF-REFINE generates an initial output, provides feedback on the output, and refines the output according to the feedback. SELF-REFINE iterates between feedback and refinement until a desired condition is met. SELF-REFINE relies on a suitable language model and three prompts (for initial generation, feedback, and refinement), and does not require training. SELF-REFINE is shown in Figure 1 and Algorithm 1. Next, we describe SELF-REFINE in more detail.

**Initial generation** Given an input $x$, prompt $p_{\text{gen}}$, and model $\mathcal{M}$, SELF-REFINE generates an initial output $y_0$:

$$y_0 = \mathcal{M}\left(p_{\text{gen}}\|x\right). \tag{1}$$

| (a) **Dialogue:** $x$, $\mathbf{y_t}$ | (b) FEEDBACK **fb** | (c) REFINE $\mathbf{y_{t+1}}$ |
|---|---|---|

```
User: I am interested
in playing Table
tennis.

Response: I'm sure
it's a great way to
socialize, stay active
```

```
Engaging: Provides no
information about table
tennis or how to play it.

User understanding: Lacks
understanding of user's
needs and state of mind.
```

```
Response (refined): That's
great to hear (...) ! It's
a fun sport requiring
quick reflexes and good
hand-eye coordination.
Have you played before, or
are you looking to learn?
```

| (d) **Code optimization:** $x$, $\mathbf{y_t}$ | (e) FEEDBACK **fb** | (f) REFINE $\mathbf{y_{t+1}}$ |
|---|---|---|

```
Generate sum of 1, ..., N
def sum(n):
    res = 0
    for i in range(n+1):
        res += i
    return res
```

```
This code is slow as
it uses brute force.
A better approach is
to use the formula
... (n(n+1))/2.
```

```
Code (refined)

def sum_faster(n):
    return (n*(n+1))//2
```

Figure 2: Examples of SELF-REFINE: an initial output ▨ generated by the base LLM and then passed back to the *same* LLM to receive feedback ▨ to the *same* LLM to refine the output ▨. The top row illustrates this for dialog generation where an initial dialogue response can be transformed into a more engaging one that also understands the user by applying feedback. The bottom row illustrates this for code optimization where the code is made more efficient by applying feedback.

---

**Algorithm 1** SELF-REFINE algorithm

---

**Require:** input $x$, model $\mathcal{M}$, prompts $\{p_{\text{gen}}, p_{\text{fb}}, p_{\text{refine}}\}$, stop condition $\text{stop}(\cdot)$
1: $y_0 = \mathcal{M}(p_{\text{gen}} \| x)$ ▷ Initial generation (Eqn. 1)
2: **for** iteration t $\in 0, 1, \ldots$ **do**
3: $\quad fb_t = \mathcal{M}(p_{\text{fb}} \| x \| y_t)$ ▷ Feedback (Eqn. 2)
4: $\quad$ **if** $\text{stop}(fb_t, t)$ **then** ▷ Stop condition
5: $\quad\quad$ break
6: $\quad$ **else**
7: $\quad\quad y_{t+1} = \mathcal{M}(p_{\text{refine}} \| x \| y_0 \| fb_0 \| \ldots \| y_t \| fb_t)$ ▷ Refine (Eqn. 4)
8: $\quad$ **end if**
9: **end for**
10: **return** $y_t$

---

Figure 3: The SELF-REFINE algorithm. See (§2) for a discussion of each component.

For example, in Figure 2(d), the model generates functionally correct code for the given input. Here, $p_{\text{gen}}$ is a task-specific few-shot prompt (or instruction) for an initial generation, and $\|$ denotes concatenation. The few-shot prompt contains input-output pairs $\langle x^{(k)}, y^{(k)} \rangle$ for the task.[3]

**FEEDBACK** Next, SELF-REFINE uses the same model $\mathcal{M}$ to provide feedback $fb_t$ on its own output, given a task-specific prompt $p_{\text{fb}}$ for generating feedback:

$$fb_t = \mathcal{M}(p_{\text{fb}} \| x \| y_t). \tag{2}$$

Intuitively, the feedback may address multiple aspects of the output. For example, in code optimization, the feedback might address the efficiency, readability, and overall quality of the code.

---

[3]Few-shot prompting (also referred to as "in-context learning") provides a model with a prompt consisting of $k$ in-context examples of the target task, each in the form of input-output pairs $\langle x_i, y_i \rangle$ (Brown et al., 2020).

Here, the prompt $p_{\text{fb}}$ provides examples of feedback in the form of input-output-feedback triples $\langle x^{(k)}, y^{(k)}, fb^{(k)} \rangle$. We prompt the model to write feedback that is actionable and specific via $fb^{(k)}$. By 'actionable', we mean the feedback should contain a concrete action that would likely improve the output. By 'specific', we mean the feedback should identify concrete phrases in the output to change. For example, the feedback in Figure 2(e) is "*This code is slow as it uses a for loop which is brute force. A better approach is to use the formula ...* `(n(n+1))/2`". This feedback is actionable, since it suggests the action 'use the formula...'. The feedback is specific since it mentions the 'for loop'.

**REFINE**    Next, SELF-REFINE uses $\mathcal{M}$ to refine its most recent output, given its own feedback:

$$y_{t+1} = \mathcal{M}\left(p_{\text{refine}} \| x \| y_t \| fb_t\right). \tag{3}$$

For example, in Figure 2(f), given the initial output and the generated feedback, the model generates a re-implementation that is shorter and runs much faster than the initial implementation. The prompt $p_{\text{refine}}$ provides examples of improving the output based on the feedback, in the form of input-output-feedback-refined quadruples $\langle x^{(k)}, y_t^{(k)}, fb_t^{(k)}, y_{t+1}^{(k)} \rangle$.

**Iterating SELF-REFINE** SELF-REFINE alternates between FEEDBACK and REFINE steps until a stopping condition is met. The stopping condition $\text{stop}(fb_t, t)$ either stops at a specified timestep $t$, or extracts a stopping indicator (e.g. a scalar stop score) from the feedback. In practice, the model can be prompted to generate a stopping indicator in $p_{\text{fb}}$, and the condition is determined per-task.

To inform the model about the previous iterations, we retain the history of previous feedback and outputs by appending them to the prompt. Intuitively, this allows the model to learn from past mistakes and avoid repeating them. More precisely, Equation (3) is in fact instantiated as:

$$y_{t+1} = \mathcal{M}\left(p_{\text{refine}} \| x \| y_0 \| fb_0 \| ... \| y_t \| fb_t\right). \tag{4}$$

Finally, we use the last refinement $y_t$ as the output of SELF-REFINE.

Algorithm 1 summarizes SELF-REFINE, and Figure 2 shows an example of SELF-REFINE in the Dialogue Response Generation (Mehri and Eskenazi, 2020) and Code Optimization (Madaan et al., 2023) tasks. Appendix V provides examples of the $p_{\text{gen}}, p_{\text{fb}}, p_{\text{refine}}$ prompts for various tasks. The key idea is that SELF-REFINE uses the same underlying LLM to generate, get feedback, and refine its outputs given its own feedback. It relies only on supervision present in the few-shot examples.

# 3   Evaluation

We evaluate SELF-REFINE on 7 diverse tasks: Dialogue Response Generation (Appendix P; Mehri and Eskenazi, 2020), Code Optimization (Appendix Q; Madaan et al., 2023), Code Readability Improvement (Appendix O; Puri et al., 2021), Math Reasoning (Appendix R; Cobbe et al., 2021), Sentiment Reversal (Appendix S; Zhang et al., 2015), and we introduce two new tasks: Acronym Generation (Appendix T) and Constrained Generation (a harder version of Lin et al. (2020) with 20-30 keyword constraints instead of 3-5; Appendix U)

Examples for all tasks and dataset statistics are provided in Table 4 (Appendix A).

## 3.1   Instantiating SELF-REFINE

We instantiate SELF-REFINE following the high-level description in Section 2. The FEEDBACK-REFINE iterations continue until the desired output quality or task-specific criterion is reached, up to a maximum of 4 iterations. To make our evaluation consistent across different models, we implemented both FEEDBACK and REFINE as few-shot prompts even with models that respond well to instructions, such as CHATGPT and GPT-4.

**Base LLMs**    Our main goal is to evaluate whether we can improve the performance of any strong base LLMs using SELF-REFINE. Therefore, we compare SELF-REFINE to the same base LLMs but without feedback-refine iterations. We used three main strong base LLM across all tasks: GPT-3.5 (`text-davinci-003`), CHATGPT (`gpt-3.5-turbo`), and GPT-4 (OpenAI, 2023). For code-based tasks, we also experimented with CODEX (`code-davinci-002`). In all tasks, either GPT-3.5 or GPT-4 is the previous state-of-the-art.[4] We used the same prompts from previous work when available

---

[4]A comparison with other few-shot and fine-tuned approaches is provided in Appendix H

| Task | GPT-3.5 Base | GPT-3.5 +Self-Refine | ChatGPT Base | ChatGPT +Self-Refine | GPT-4 Base | GPT-4 +Self-Refine |
|---|---|---|---|---|---|---|
| Sentiment Reversal | 8.8 | **30.4** (↑21.6) | 11.4 | **43.2** (↑31.8) | 3.8 | **36.2** (↑32.4) |
| Dialogue Response | 36.4 | **63.6** (↑27.2) | 40.1 | **59.9** (↑19.8) | 25.4 | **74.6** (↑49.2) |
| Code Optimization | 14.8 | **23.0** (↑8.2) | 23.9 | **27.5** (↑3.6) | 27.3 | **36.0** (↑8.7) |
| Code Readability | 37.4 | **51.3** (↑13.9) | 27.7 | **63.1** (↑35.4) | 27.4 | **56.2** (↑28.8) |
| Math Reasoning | **64.1** | **64.1** (0) | 74.8 | **75.0** (↑0.2) | 92.9 | **93.1** (↑0.2) |
| Acronym Generation | 41.6 | **56.4** (↑14.8) | 27.2 | **37.2** (↑10.0) | 30.4 | **56.0** (↑25.6) |
| Constrained Generation | 16.0 | **39.7** (↑23.7) | 2.75 | **33.5** (↑30.7) | 4.4 | **61.3** (↑56.9) |

Table 1: SELF-REFINE results on various tasks using GPT-3.5, CHATGPT, and GPT-4 as base LLM. SELF-REFINE consistently improves LLM. Metrics used for these tasks are defined in Section 3.2.

(such as for Code Optimization and Math Reasoning); otherwise, we created prompts as detailed in Appendix V. We generate samples using a temperature of 0.7.

## 3.2 Metrics

We report three types of metrics:

- Task specific metric: When available, we use automated metrics from prior work (Math Reasoning: % solve rate; Code Optimization: % programs optimized%).

- GPT-4-pref: In addition to human-pref, we use GPT-4 as a proxy for human preference following prior work (Fu et al., 2023; Chiang et al., 2023; Geng et al., 2023; Sun et al., 2023), and found high correlation (82% for Sentiment Reversal, 68% for Acronym Generation, and 71% for Dialogue Response Generation) with human-pref. For Code Readability Improvement, we prompt GPT-4 to calculate fraction of the variables that are appropriately named given the context (e.g., x = [] → input_buffer = []). Additional details are provided in Appendix F. For constrained generation, we combine automated evaluation to quantify concept coverage and GPT-4-pref to ensure the commonsense correctness of generated sentences. A sentence is only deemed a winner if it maintains validity in commonsense reasoning and has greater coverage in terms of concepts.

- Human evaluation: In Dialogue Response Generation, Code Readability Improvement, Sentiment Reversal, and Acronym Generation, we additionally perform a blind human A/B evaluation on a subset of the outputs to select the preferred output. Additional details are provided in Appendix C.

## 3.3 Results

Table 1 shows our main results:

**SELF-REFINE consistently improves over base models** across all model sizes, and additionally outperforms the previous state-of-the-art across all tasks. For example, GPT-4+SELF-REFINE improves over the base GPT-4 by 8.7% (absolute) in Code Optimization, increasing optimization percentage from 27.3% to 36.0%. Confidence intervals are provided in Appendix M. For code-based tasks, we found similar trends when using CODEX; those results are included in Appendix H.

One of the tasks in which we observe the highest gains compared to the base models is Constrained Generation, where the model is asked to generate a sentence containing up to 30 given concepts. We believe that this task benefits significantly from SELF-REFINE because there are more opportunities to miss some of the concepts on the first attempt, and thus SELF-REFINE allows the model to fix these mistakes subsequently. Further, this task has an extremely large number of reasonable outputs, and thus SELF-REFINE allows to better explore the space of possible outputs.

In preference-based tasks such as Dialogue Response Generation, Sentiment Reversal, and Acronym Generation, SELF-REFINE leads to especially high gains. For example in Dialogue Response Generation, GPT-4 preference score improve by 49.2% – from 25.4% to 74.6%. Similarly, we see remarkable improvements in the other preference-based tasks across all models.

The modest performance gains in Math Reasoning can be traced back to the inability to accurately identify whether there is any error. In math, errors can be nuanced and sometimes limited to a single line or incorrect operation. Besides, a consistent-looking reasoning chain can deceive LLMs to think that "everything looks good" (e.g., CHATGPT feedback for 94% instances is 'everything looks good'). In Appendix K.1, we show that the gains with SELF-REFINE on Math Reasoning are much bigger (5%+) if an external source can identify if the current math answer is incorrect. Although SELF-REFINE demonstrates limited efficacy in Math Reasoning, we observe gains with SELF-REFINE in a subset of Big-Bench Hard (Suzgun et al., 2022) tasks that typically require a combination of commonsense reasoning and logic, such as date reasoning (Appendix D). This suggests that SELF-REFINE may be more effective in scenarios where the interplay of logical analysis and knowledge acquired through pre-training facilitates self-verification.

**Improvement is consistent across base LLMs sizes** Generally, GPT-4+SELF-REFINE performs better than GPT-3.5+SELF-REFINE and CHATGPT+SELF-REFINE across all tasks, even in tasks where the initial base results of GPT-4 were lower than GPT-3.5 or CHATGPT. We thus believe that SELF-REFINE allows stronger models (such as GPT-4) to unlock their full potential, even in cases where this potential is not expressed in the standard, single-pass, output generation. Comparison to additional strong baselines is provided in Appendix H.

## 4 Analysis

The three main steps of SELF-REFINE are FEEDBACK, REFINE, and repeating them iteratively. In this section, we perform additional experiments to analyze the importance of each of these steps.

| Task | SELF-REFINE feedback | Generic feedback | No feedback |
|---|---|---|---|
| Code Optimization | **27.5** | 26.0 | 24.8 |
| Sentiment Reversal | **43.2** | 31.2 | 0 |
| Acronym Generation | **56.4** | 54.0 | 48.0 |

Table 2: Prompting to generate generic feedback (or having the model generate no feedback at all) leads to reduced scores, indicating the importance of the FEEDBACK step of SELF-REFINE. These experiments were performed with CHATGPT (Code Optimization and Sentiment Reversal) and GPT-3.5 (Acronym Generation), and metrics used are defined in Section 3.2.

**The impact of the feedback quality** Feedback quality plays a crucial role in SELF-REFINE. To quantify its impact, we compare SELF-REFINE, which utilizes specific, actionable feedback, with two ablations: one using generic feedback and another without feedback (the model may still iteratively refine its generations, but is not explicitly provided feedback to do so). For example, in the Code Optimization task: actionable feedback, such as *Avoid repeated calculations in the for loop*, pinpoints an issue and suggests a clear improvement. Generic feedback, like *Improve the efficiency of the code*, lacks this precision and direction. Table 2 shows feedback's clear influence.

In Code Optimization, performance slightly dips from 27.5 (SELF-REFINE feedback) to 26.0 (generic feedback), and further to 24.8 (no feedback). This suggests that while generic feedback offers some guidance – specific, actionable feedback yields superior results.

This effect is more pronounced in tasks like Sentiment Reversal, where changing from our feedback to generic feedback leads to a significant performance drop (43.2 to 31.2), and the task fails without feedback. In the "No feedback" setting, the model was not given clear instructions on changing the output. We find that the model tends to either repeat the same output in each iteration or to make unrelated changes. Since the scores in this task are the relative improvement increase in human preference, a score of 0 means that "No feedback" did not improve over the base model outputs. Similarly, in Acronym Generation, without actionable feedback, performance drops from 56.4 to 48.0, even with iterative refinements. These results highlight the importance of specific, actionable feedback in our approach. Even generic feedback provides some benefit, but the best results are achieved with targeted, constructive feedback.

**How important are the multiple iterations of FEEDBACK-REFINE?** Figure 4 demonstrates that on average, the quality of the output improves as the number of iterations increases. For instance, in

| Task | $y_0$ | $y_1$ | $y_2$ | $y_3$ |
|------|------|------|------|------|
| C. Opt. | 22.0 | 27.0 | 27.9 | **28.8** |
| S. Rev. | 33.9 | 34.9 | 36.1 | **36.8** |
| C. Gen. | 12.1 | 26.1 | 39.6 | **46.1** |

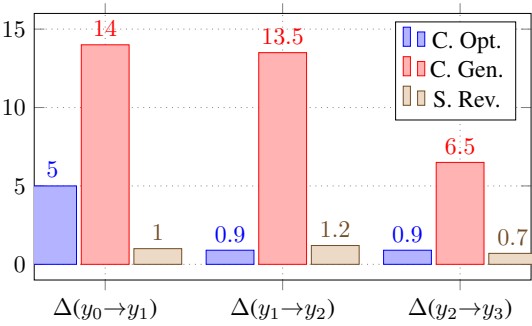

Figure 4: **Left**: Iteration-wise score improvements. Early iterations significantly improve output quality, and scores generally keep improving with more iterations. **Right**: SELF-REFINE Performance improvements with iterations. Most gains($\Delta$) are in the initial iterations for both Code Opt. and Sentiment Reversal. The numbers are averaged over CHATGPT, GPT-3.5, and GPT-4. Task abbreviations: C. Opt. (Code Optimization), S. Rev. (Sentiment Reversal), C. Gen. (Constrained Generation).

```python
# Slower code
def solve(amount):
  best_price = (amount + 199) // 200 *
  ↪   380
  # First loop
  for a in range(amount // 200 + 1):
    # ... 4 nested loops ...
    for c1 in range(amount // 1500 +
    ↪   1):
      if a*200 + b*300 == amount:
        price = a*380 + b*550
        if price < best_price:
          best_price = price
  return best_price
```

```python
# Faster code
def solve(amount):
  coins = [200, 300]
  prices = [380, 550]
  dp = [float('inf')] * (amount + 1)
  dp[0] = 0
  for i in range(len(coins)):
    for j in range(coins[i], amount+1):
      dp[j] = min(dp[j], dp[j -
      ↪   coins[i]] + prices[i])
  return dp[amount]
```

Figure 5: Comparison of code generated by Madaan et al. (2023) (left) and the output after applying SELF-REFINE (right). The initial code by the baseline, which is nearly identical to the slower input program, fails to improve the efficiency and merely alters the logic for reading input. SELF-REFINE first generates feedback that diagnoses that *This code is slow because it is using six nested loops to iterate through all possible combinations of coins to pay the amount*, and suggests that *a more efficient approach would be ...*. SELF-REFINE then uses this feedback to generate the revised code (right), reducing the time complexity to $\mathcal{O}(amount * coins)$. The full example is provided in Appendix K

the Code Optimization task, the initial output ($y_0$) has a score of 22.0, which improves to 28.8 after three iterations ($y_3$). Similarly, in the Sentiment Reversal task, the initial output has a score of 33.9, which increases to 36.8 after three iterations. This trend of improvement is also evident in Constrained Generation, where the score increases from 26.1 to 46.1 after three iterations. Figure 4 highlights the diminishing returns in the improvement as the number of iterations increases. Overall, having multiple FEEDBACK-REFINE iterations significantly enhances the quality of the output, although the marginal improvement naturally decreases with more iterations.

The performance may not always monotonically increase with iterations: in multi-aspect feedback tasks like Acronym Generation, where the output quality can vary during iteration with improvement

in one aspect but decline in another aspect. To counter this, SELF-REFINE generates numerical scores for different quality aspects, leading to a balanced evaluation and appropriate output selection.

**Can we just generate multiple outputs instead of refining?**    Does SELF-REFINE improve because of the iterative refinement, or just because it generates *more* outputs? We compare SELF-REFINE with CHATGPT, when CHATGPT generates $k = 4$ samples (but without feedback and refinement). Then, we compare the performance of SELF-REFINE against these $k$ initial outputs in a 1 vs. $k$ evaluation. In other words, we assess whether SELF-REFINE can outperform *all* $k$ initial outputs. The results of this experiment are illustrated in Figure 7 (Appendix K). Despite the increased difficulty of the 1 vs. $k$ setting, the outputs of SELF-REFINE are still preferred by humans over *all* $k$ initial outputs. This shows the importance of refinement according to feedback over the alternative of just generating multiple initial outputs.

**Does SELF-REFINE works in an instruction only setup?**    In our main experiments, we use few-shot prompting to guide model output into a more readily parseable format. Next, we experiment with SELF-REFINE under a zero-shot prompting scenario, where traditional few-shot examples are supplanted by explicit instructions at each stage of the SELF-REFINE process. For these experiments, we use CHATGPT. The results (Appendix E in Table 8) show that SELF-REFINE remains effective across diverse tasks, even in the absence of example prompts. Notably, in tasks such as Acronym Generation and Sentiment Reversal, SELF-REFINE, under zero-shot prompting, enhances performance from 16.6% to 44.8% and 4.4% to 71.4%, respectively. However, achieving optimal performance in this setting requires extensive prompt engineering for instructions.

For Math Reasoning tasks, SELF-REFINE improves the solve rate from 22.1% to 59.0% in an instruction-only setting. We find that much of this gain comes from fixing omitted return statements in 71% of the initial Python programs, despite clear instructions to include them. Subsequent iterations of feedback generation and refinement address this issue effectively, decreasing the error rate by 19%. Further, we find that when the initial programs are valid, SELF-REFINE does not improve the performance.

**Does SELF-REFINE work with weaker models?**    The experiments in Section 3.3 were performed with some of the strongest available models; does SELF-REFINE work with smaller or weaker models as well? To investigate this, we instantiated SELF-REFINE with Vicuna-13B (Chiang et al., 2023), a less powerful base model. While Vicuna-13B is capable of generating initial outputs, it struggles significantly with the refinement process. Specifically, Vicuna-13B was not able to consistently generate the feedback in the required format. Furthermore, even when provided with Oracle or hard-coded feedback, it often failed to adhere to the prompts for refinement. Instead of refining its output, Vicuna-13B either repeated the same output or generated a hallucinated conversation, rendering the outputs less effective. Example output and analysis is provided in Appendix I.

**How does SELF-REFINE perform with strong open access models like LLAMA2-70B?**    We conduct additional experiments using SELF-REFINE on LLama-2 (Touvron et al., 2023), a state-of-the-art, open-access language model on Acronym Generation, Sentiment Reversal, Dialogue Response Generation, and Math Reasoning. Consistent with our primary findings, SELF-REFINE shows an improvement across all these tasks relative to the base model. The full results are shown in Appendix J. These promising results with LLAMA2-70B suggest that the applicability of SELF-REFINE might extend to a wide array of increasingly powerful open-source models in the future

**Qualitative Analysis**    We conduct a qualitative analysis of the feedback generated by SELF-REFINE and its subsequent refinements. We manually analyze 70 samples in total (35 success cases and 35 failure cases) for Code Optimization (Madaan et al., 2023) and Math Reasoning (Cobbe et al., 2021). For both Math Reasoning and Code Optimization, we found that the feedback was predominantly actionable, with the majority identifying problematic aspects of the original generation and suggesting ways to rectify them.

When SELF-REFINE failed to improve the original generation, the majority of issues were due to erroneous feedback rather than faulty refinements. Specifically, 33% of unsuccessful cases were due to feedback inaccurately pinpointing the error's location, while 61% were a result of feedback suggesting an inappropriate fix. Only 6% of failures were due to the refiner incorrectly implementing good feedback. These observations highlight the vital role of accurate feedback plays in SELF-REFINE.

| | Supervision-free refiner | Supervision-free feedback | Multi-aspect feedback | Iterative |
|---|---|---|---|---|
| **Learned refiners**: PEER (Schick et al., 2022b), Self-critique (Saunders et al., 2022b), CodeRL (Le et al., 2022b), Self-correction (Welleck et al., 2022). | ❌ | ✅ or ❌ | ❌ | ✅ or ❌ |
| **Prompted refiners**: Augmenter (Peng et al., 2023), Re³ (Yang et al., 2022), Reflexion (Shinn et al., 2023). | ✅ | ✅ or ❌ | ❌ | ❌ |
| SELF-REFINE (this work) | ✅ | ✅ | ✅ | ✅ |

Table 3: A comparison of SELF-REFINE to closely related prior refinement approaches.

In successful cases, the refiner was guided by accurate and useful feedback to make precise fixes to the original generation in 61% of the cases. Interestingly, the refiner was capable of rectifying issues even when the feedback was partially incorrect, which was the situation in 33% of successful cases. This suggests resilience to sub-optimal feedback. Future research could focus on examining the refiner's robustness to various types of feedback errors and exploring ways to enhance this resilience. In Figure 5, we illustrate how SELF-REFINE significantly improves program efficiency by transforming a brute force approach into a dynamic programming solution, as a result of insightful feedback. Additional analysis on other datasets such as Dialogue Response Generation is provided in Appendix K.

**Going Beyond Benchmarks** While our evaluation focuses on benchmark tasks, SELF-REFINE is designed with broader applicability in mind. We explore this in a real-world use case of website generation, where the user provides a high-level goal and SELF-REFINE assists in iteratively developing the website. Starting from a rudimentary initial design, SELF-REFINE refines HTML, CSS, and JS to evolve the website in terms of both usability and aesthetics. This demonstrates the potential of SELF-REFINE in real-world, complex, and creative tasks. See Appendix L for examples and further discussion, including broader, societal impact of our work.

## 5 Related work

Leveraging human- and machine-generated natural language (NL) feedback for refining outputs has been effective for a variety of tasks, including summarization (Scheurer et al., 2022), script generation (Tandon et al., 2021), program synthesis (Le et al., 2022a; Yasunaga and Liang, 2020), and other tasks (Madaan et al., 2022; Bai et al., 2022a; Schick et al., 2022b; Saunders et al., 2022a; Bai et al., 2022b; Welleck et al., 2022). Refinement methods differ in the source and format of feedback, and the way that a refiner is obtained. Table 3 summarizes some related approaches; see Appendix B for an additional discussion.

**Source of feedback.** Humans have been an effective source of feedback (Tandon et al., 2021; Elgohary et al., 2021; Tandon et al., 2022; Bai et al., 2022a). Since human feedback is costly, several approaches use a scalar reward function as a surrogate of (or alternative to) human feedback (e.g., (Bai et al., 2022a; Liu et al., 2022; Lu et al., 2022; Le et al., 2022a; Welleck et al., 2022)). Alternative sources such as compilers (Yasunaga and Liang, 2020) or Wikipedia edits (Schick et al., 2022b) can provide domain-specific feedback. Recently, LLMs have been used to generate feedback for general domains (Fu et al., 2023; Peng et al., 2023; Yang et al., 2022), However, ours is the only method that generates feedback using an LLM on its *own* output, for the purpose of refining with the same LLM.

**Representation of feedback.** The form of feedback can be generally divided into natural language (NL) and non-NL feedback. Non-NL feedback can come in human-provided example pairs (Dasgupta et al., 2019) or scalar rewards (Liu et al., 2022; Le et al., 2022b). In this work, we use NL feedback, since this allows the model to easily provide *self*-feedback using the same LM that generated the output, while leveraging existing pretrained LLMs such as GPT-4.

**Types of refiners.** Pairs of feedback and refinement have been used to learn supervised refiners (Schick et al., 2022b; Du et al., 2022; Yasunaga and Liang, 2020; Madaan et al., 2021). Since

gathering supervised data is costly, some methods learn refiners using model generations (Welleck et al., 2022; Peng et al., 2023). However, the refiners are trained for each new domain. Finally, (Yang et al., 2022) use prompted feedback and refinement specifically tailored for story generation. In this work, we avoid training a separate refiner, and show that the same model can be used as both the refiner and the source of feedback across multiple domains.

**Non-refinement reinforcement learning (RL) approaches.** Rather than having explicit refinement, an alternative way to incorporate feedback is by optimizing a scalar reward function, e.g. with reinforcement learning (e.g., Stiennon et al. (2020); Lu et al. (2022); Le et al. (2022a)). These methods differ from SELF-REFINE in that the model does not access feedback on an intermediate generation. Second, these RL methods require updating the model's parameters, unlike SELF-REFINE. Recently, in discrete-space simulated environments, LLMs have also been shown to iteratively shape and refine rewards and policies, thereby performing RL tasks without expert demonstrations or gradients (Kim et al., 2023; Brooks et al., 2023). While we focus on real-world code and language tasks in this paper, it would be interesting to explore applications of self-refine in simulated environments.

# 6 Limitations and Discussion

The main limitation of our approach is that the base models need to have sufficient few-shot modeling or instruction-following abilities, in order to learn to provide feedback and to refine in an in-context fashion, without having to train supervised models and rely on supervised data.

Further, the experiments in this work were primarily performed with language models that are not open-sourced, namely GPT-3.5, CHATGPT, GPT-4, and CODEX. Existing literature (Ouyang et al., 2022) does not fully describe the details of these models, such as the pretraining corpus, model sizes, and model biases. Nonetheless, we release our code and model outputs to ensure the reproducibility of our work. In addition, initial results from our experiments with the open-access LLAMA2-70B language model are promising, reinforcing the notion that SELF-REFINE has the potential to be widely applicable, even as open-source models continue to evolve and improve.

Another limitation of our work is that we exclusively experiment with datasets in English. In other languages, the current models may not provide the same benefits. Finally, there is a possibility for bad actors to use prompting techniques to steer a model to generate more toxic or harmful text. Our approach does not explicitly guard against this.

# 7 Conclusion

We present SELF-REFINE: a novel approach that allows large language models to iteratively provide self-feedback and refine their own outputs. SELF-REFINE operates within a single LLM, requiring neither additional training data nor reinforcement learning. We demonstrate the simplicity and ease of use of SELF-REFINE across a wide variety of tasks. By showcasing the potential of SELF-REFINE in diverse tasks, our research contributes to the ongoing exploration and development of large language models, with the aim of reducing the cost of human creative processes in real-world settings. We hope that our iterative approach will help drive further research in this area. To this end, we make all our code, data and prompts available at https://selfrefine.info/.

## Acknowledgements

We thank the anonymous reviewers for their useful comments and suggestions, and extend our thanks to James Laudon, Cliff Young, Graham Neubig, Daniel Fried, Rif A. Saurous, and the Google DeepMind team for their valuable feedback and comments. This material is partly based on research sponsored in part by the Air Force Research Laboratory (agreement number FA8750-19-2-0200). The U.S. Government is authorized to reproduce and distribute reprints for Governmental purposes notwithstanding any copyright notation thereon. The views and conclusions contained herein are those of the authors and should not be interpreted as necessarily representing the official policies or endorsements, either expressed or implied, of DARPA or the U.S. Government.

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

# A Evaluation Tasks

Table 4 lists the tasks in our evaluation, and examples from each task.

| Task and Description | Sample one iteration of FEEDBACK-REFINE |
|---|---|
| **Sentiment Reversal**
Rewrite reviews to reverse sentiment.
Dataset: (Zhang et al., 2015) 1000 review passages | $x$: The food was fantastic..."
$y_t$: The food was disappointing..."
$fb$: Increase negative sentiment
$y_{t+1}$: The food was utterly terrible..." |
| **Dialogue Response Generation**
Produce rich conversational responses.
Dataset: (Mehri and Eskenazi, 2020) 372 conv. | $x$: What's the best way to cook pasta?"
$y_t$: The best way to cook pasta is to..."
$fb$: Make response relevant, engaging, safe
$y_{t+1}$: Boil water, add salt, and cook pasta..." |
| **Code Optimization**
Enhance Python code efficiency
Dataset: (Madaan et al., 2023): 1000 programs | $x$: Nested loop for matrix product
$y_t$: NumPy dot product function
$fb$: Improve time complexity
$y_{t+1}$: Use NumPy's optimized matmul function |
| **Code Readability Improvement**
Refactor Python code for readability.
Dataset: (Puri et al., 2021) 300 programs* | $x$: Unclear variable names, no comments
$y_t$: Descriptive names, comments
$fb$: Enhance variable naming; add comments
$y_{t+1}$: Clear variables, meaningful comments |
| **Math Reasoning**
Solve math reasoning problems.
Dataset: (Cobbe et al., 2021) 1319 questions | $x$: Olivia has \$23, buys 5 bagels at \$3 each"
$y_t$: Solution in Python
$fb$: Show step-by-step solution
$y_{t+1}$: Solution with detailed explanation |
| **Acronym Generation**
Generate acronyms for a given title
Dataset: (Appendix T) 250 acronyms | $x$ : Radio Detecting and Ranging"
$y_t$: RDR
$fb$ : be context relevant; easy pronunciation
$y_{t+1}$: RADAR" |
| **Constrained Generation**
Generate sentences with given keywords.
Dataset: (Lin et al., 2020) 200 samples | $x$: beach, vacation, relaxation
$y_t$: During our beach vacation...
$fb$: Include keywords; maintain coherence
$y_{t+1}$: .. beach vacation was filled with relaxation |

Table 4: An overview of the tasks which we evaluate SELF-REFINE on, along with their associated datasets and sizes. For every task, we demonstrate a single iteration of refinement of input $x$, the previously generated output $y_t$, the feedback generated $fb_t$, and the refinement $y_{t+1}$. Few-shot prompts used for FEEDBACK and REFINE are provided in Appendix V.

# B   Broader Related Work

Compared to a concurrent work, Reflexion (Shinn et al., 2023), our approach involves correction using feedback, whereas their setup involves finding the next best solution in planning using ReAct. While ReAct and Reflexion provide a free-form reflection on whether a step was executed correctly and potential improvements, our approach is more granular and structured, with multi-dimensional feedback and scores. This distinction allows our method to offer more precise and actionable feedback, making it suitable for a wider range of natural language generation tasks, including those that may not necessarily involve step-by-step planning such as open-ended dialogue generation.

**Comparison with Welleck et al. (2022)**   The closest work to ours may be Self-Correction (Welleck et al., 2022); however, Self-Correction has several disadvantages compared to SELF-REFINE:

1. Self-Correction does not train their model to generate explicit feedback; instead, Welleck et al. (2022) trained their models to refine only. As we show in Section 4 and Table 2, having the model generate explicit feedback results in significantly better refined outputs.

2. Self-Correction trains a separate refiner (or "corrector") for each task. In contrast, SELF-REFINE uses instructions and few-shot prompting, and thus does not require training a separate refiner for each task.

3. Empirically, we evaluated SELF-REFINE using the same base model of GPT-3 as Self-Correction, and with the same settings on the GSM8K benchmark. Self-Correction achieved 45.9% accuracy while SELF-REFINE (this work) achieved **55.7%** (↑**9.8**).

**Comparison with non-refinement reinforcement learning (RL) approaches.**   Rather than having an explicit refinement module, an alternative way to incorporate feedback is by optimizing a scalar reward function, e.g. with reinforcement learning (e.g., Stiennon et al. (2020); Lu et al. (2022); Le et al. (2022a)). These methods differ from SELF-REFINE (and more generally, refinement-based approaches) in that the model cannot access feedback on an intermediate generation. Second, these reinforcement learning methods require updating the model's parameters, unlike SELF-REFINE.

See Table 5 for an additional detailed comparison of related work.

| Method | Primary Novelty | zero/few shot improvement | multi aspect critics | NL feedback with error localization | iterative framework |
|---|---|---|---|---|---|
| RLHF (Stiennon et al., 2020) | optimize for human preference | ✗trained on feedback | ✗single (human) | ✓(not self gen.) | ✗ |
| Rainier RL (Liu et al., 2022) | RL to generate knowledge | ✗trained on end task | ✗single(accuracy) | ✗(knowl. only) | ✗ |
| QUARK RL (Lu et al., 2022) | quantization to edit generations | ✗trained on end task | ✗single(scalar score) | ✗(dense signal) | ✓(train time iter.) |
| Code RL (Le et al., 2022a) | actor critic RL for code improvement | ✗trained on end task | ✗single(unit tests) | ✗(dense signal) | ✗ |
| DrRepair (Yasunaga and Liang, 2020) | Compiler feedback to iteratively repair | ✗trained semi sup. | ✗single(compiler msg) | ✓(not self gen.) | ✓ |
| PEER (Schick et al., 2022b) | doc. edit trained on wiki edits | ✗trained on edits | ✗single(accuracy) | ✓(not self gen.) | ✓ |
| Self critique (Saunders et al., 2022a) | few shot critique generation | ✗feedback training | ✗single(human) | ✓(self gen.) | ✗ |
| Self-correct (Welleck et al., 2022) | novel training of a corrector | ✗trained on end task | ✗single (task specific) | ✓(limited setting) | ✓(limited setting) |
| Const. AI (Bai et al., 2022b) | train RL4F on automat (critique, revision) pair | ✗critique training | ✓(fixed set) | ✓ | ✗ |
| Self-ask (Press et al., 2022) | ask followup ques when interim ans correct;final wrong | ✓few shot | ✗none | ✗(none) | ✗ |
| GPT3 score (Fu et al., 2023) | GPT can score generations with instruction | ✓few shot | ✗single(single utility fn) | ✗(none) | ✗ |
| Augmenter (Peng et al., 2023) | factuality feedback from external KBs | ✓few shot | ✗single(factuality) | ✓(self gen.) | ✓ |
| Re³ (Yang et al., 2022) | ~ours: but one domain, trained critics | ✓few shot | ✓(trained critics) | ✓(not self gen.) | ✓ |
| SELF-REFINE | fewshot iterative multi aspect NL fb | ✓few shot | ✓multiple(few shot critics) | ✓(self gen.) | ✓ |

Table 5: Summary of related approaches. Reinforcement learning approaches are shown in   purple   , trained corrector approaches are shown in   orange   , and few-shot corrector approaches are shown in   green   .

# C   Human Evaluation

The A/B evaluation in our study was conducted by the authors, where a human judge was presented with an input, task instruction, and two candidate outputs generated by the baseline method and SELF-REFINE. The setup was blind, i.e., the judges did not know which outputs were generated by which method. The judge was then asked to select the output that is better aligned with the task instruction. For tasks that involve A/B evaluation, we calculate the relative improvement as the percentage increase in preference rate. The preference rate represents the proportion of times annotators selected the output produced by SELF-REFINE over the output from the baseline method. Table 6 shows the results.

While multiple annotators participated in each task, we only collected a single annotation per instance, aiming to scale the number of data points we could annotate. To further validate the reliability of our evaluations, we obtained two annotations for 50 samples from each dataset. All human evaluations were executed in a double-blind manner, with the responses being randomly flipped to ensure annotator impartiality, leaving them unaware of whether a given output was from the base model or the SELF-REFINE. These additional evaluations were exclusively applied to outputs generated by GPT-4.

For each task, we measured inter-labeler agreement using Cohen's Kappa score. Code Readability Improvement and Acronym Generation both scored a substantial 0.75, Sentiment Reversal was also substantial at 0.61, while Dialogue Response Generation was moderate with a score of 0.53.

| Task | SELF-REFINE (%) | Direct (%) | Either (%) |
|---|---|---|---|
| Sentiment Reversal | 75.00 | 21.43 | 3.57 |
| Acronym Generation | 44.59 | 12.16 | 43.24 |
| Dialogue Response Generation | 47.58 | 19.66 | 32.76 |
| Code Readability Improvement | **50.0** | 3.0 | 47.0 |

Table 6: Relative improvement of SELF-REFINE in A/B evaluations across different tasks. The values represent normalized preferences, which correspond to the proportion of times the output generated by SELF-REFINE was selected as better aligned with the task instruction over the baseline method. The evaluation was conducted for 150 examples for each dataset. The judges were not aware of the method that generated each sample.

# D SELF-REFINE on Reasoning Tasks

SELF-REFINE shows limited success in Math Reasoning tasks, mainly due to its challenges in generating meaningful feedback. Nonetheless, we observe performance gains with SELF-REFINE in certain tasks from the Big-Bench Hard suite (Suzgun et al., 2022), particularly those requiring commonsense reasoning and logical thinking. These results in Table 7 suggest that SELF-REFINE is more effective in scenarios where the combination of logical reasoning and pre-trained knowledge contribute to self-verification.

| Task | Base model | +Self-Refine | Gain |
|------|-----------|--------------|------|
| Date Understanding | 62.0 | 66.8 | 4.8 |
| Geometric Shapes | 17.6 | 20.0 | 2.4 |
| Logical Deduction (seven objects) | 43.2 | 45.2 | 2.0 |
| Multi-Step Arithmetic | 61.6 | 64.0 | 2.4 |
| Tracking Shuffled Objects (seven objects) | 31.6 | 36.0 | 4.4 |

Table 7: Performance comparison between the base outputs and the SELF-REFINE enhanced model across various Big Bench Hard tasks. All experiments were conducted using the GPT-3.5-TURBO-0613 model with a temperature setting of 0.0. No task-specific prompts were utilized; all tasks employed the same instructional prompts.

# E Instruction-Only Prompting

In our main experiments, we use few-shot prompting to guide model output into a more readily parseable format. Next, we experiment with SELF-REFINE under a zero-shot prompting scenario, where traditional few-shot examples are supplanted by explicit instructions at each stage of the SELF-REFINE process. For these experiments, we use CHATGPT. The results in Table 8) show that SELF-REFINE remains effective across diverse tasks. However, achieving optimal performance in this setting requires extensive prompt engineering for instructions. The instructions are present in Listing 1 (Acronym Generation), Listing 3 (Math Reasoning), Listing 2 (Sentiment Reversal), Listing 4 (Constrained Generation), and Listing 5 (Dialogue Response Generation).

For Math Reasoning tasks, SELF-REFINE improves the solve rate from 22.1% to 59.0% in an instruction-only setting. We find that much of this gain comes from fixing omitted return statements in 71% of the initial Python programs, despite clear instructions to include them. Subsequent iterations of feedback generation and refinement address this issue effectively, decreasing the error rate by 19%. Further, we find that when the initial programs are valid, SELF-REFINE does not improve the performance.

| Task | Base | SELF-REFINE (zero-shot) |
|------|------|-------------------------|
| Acronym Generation | 16.6% | 44.8% (↑ 28.2%) |
| Constrained Generation | 4.0% | 47.0% (↑ 43.0%) |
| Sentiment Reversal | 4.4% | 71.4% (↑ 67.0%) |
| Math Reasoning | 22.1% | 59.0% (↑ 36.9%) |
| Dialogue Response Generation | 23.0% | 48.8% (↑ 25.8%) |

Table 8: Performance of SELF-REFINE with Zero-Shot Prompting

# F GPT-4 Evaluation

In light of the impressive achievements of GPT-4 in assessing and providing reasoning for complex tasks, we leverage its abilities for evaluation in SELF-REFINE. The approach involves presenting tasks to GPT-4 in a structured manner, promoting the model's deliberation on the task and generating a rationale for its decision. To mitigate order bias in our tasks, we randomly flip the order of the outputs generated after the first iteration and by SELF-REFINE before evaluation. Further, to ensure

**Listing 1** Instruction-only prompts used at various stages of the SELF-REFINE process for Acronym Generation.

```
# Init: Generate an acronym for a given title
start_chat_log = [
    {"role": "system", "content": 'You are a helpful assistant that generates
    ↪  acronyms.'},
    {"role": "user", "content": f'Generate an acronym for the title "{title}".
    ↪  Please respond in the format: "The generated acronym is: {acronym}"'}
]

# Generate Feedback: Evaluate the quality of the generated acronym
feedback_str = f"""Evaluate the acronym "{acronym}" for the title "{title}" on its
↪  ease of pronunciation, ease of spelling, relation to the title, positive
↪  connotation, and being well-known."""
start_chat_log = [
    {"role": "system", "content": 'You are an AI model that provides feedback on
    ↪  the viability and quality of acronyms.'},
    {"role": "user", "content": feedback_str}
]

# Refine: Improve the acronym based on provided feedback
start_chat_log = [
    {"role": "system", "content": 'You are a helpful assistant that can
    ↪  iteratively improve acronyms based on feedback.'},
    {"role": "user", "content": f'Improve the acronym "{acronym}" for the title
    ↪  "{title}" based on the following feedback: {feedback}. Please *always*
    ↪  respond in the format: "The improved acronym is: {acronym}"'}
]
```

**Listing 2** Instruction-only prompts used at various stages of the Self-Refine process for Sentiment Reversal.

```
# Init: Transform a negative sentiment review into a positive one
start_chat_log = [
    {"role": "system", "content": 'You are an AI language model that transforms a
    ↪  negative sentiment review into a positive one.'},
    {"role": "user", "content": f'Please transform the following negative review
    ↪  into a positive one: "{review}". Respond in the format of: "The positive
    ↪  review is: [Your Response Here]".'},
]

# Generate Feedback: Provide feedback on the transformed review
start_chat_log = [
    {"role": "system", "content": 'You are an AI model providing feedback on a
    ↪  sentiment transformed review.'},
    {"role": "user", "content": f"Why is this review not Very positive? Point out
    ↪  specific issues and give concrete suggestions. Respond in the format of:
    ↪  'Feedback: [Specific issues and suggestions]"},
]

# Refine: Improve the review based on provided feedback
start_chat_log = [
    {"role": "system", "content": 'You are an AI model that improves upon an
    ↪  existing review based on provided feedback.'},
    {"role": "user", "content": f'Please improve the sentiment of the following
    ↪  review "{sentence}" based on this feedback: "{feedback}", and make it more
    ↪  positive. Always respond in the format of: "The more positive review is:
    ↪  [Your Response Here]".'},
]
```

**Listing 3** Instruction-only prompts used at various stages of the Self-Refine process for Math Reasoning.

```
# Function to Generate an Answer
gen_sys_template = "You are a helpful assistant that responds with only a python
↪   program."
gen_user_template = """Write a python function that solves the given question and
↪   returns the result.
Always store the final result in a variable called `result`, and always include
↪   `return result` as your last statement.
# Question: {question}
# solution using Python:"""

# Function to Get Feedback on the Answer
fb_sys_template = "You are a helpful assistant that provides feedback on the
↪   correctness of python programs."
fb_user_template = """Question: {question}
{prediction}
# There may be an error in the code above because of lack of understanding of the
↪   question.
To find the error, go through semantically complete blocks of the code, and check
↪   if everything looks good.
Errors could include:
- Logical issues
- Lack of understanding of the question
- Missing `return result`
Share your evaluation in the format:
Evaluation: <your evaluation here>.
If everything looks good, return 'Evaluation: correct'"""

# Function to Fix the Error
refine_sys_template = "You are a helpful assistant that responds with only a
↪   python program."
refine_user_template = """Question: {question}
{prediction}
# There is an error in the code above. The following is the feedback on the code:
{feedback}
Fix the error in the python function given the feedback above. The python function
↪   should return the final answer."""
```

that there is no inherent bias in GPT-4 picking its own predictions, we conduct an additional study with CLAUDE-V2.

**Claude-2 as the Evaluator**    Despite the measures we took to prevent any inherent biases, GPT-4 might inherently favor self-refined outputs. To provide a comprehensive evaluation of GPT-4, we conduct an additional analysis using CLAUDE-V2, [5] serving as an independent evaluator for GPT-4 outputs. The results in Table 9 from GPT-4 as the base LLM with CLAUDE-V2 as the evaluator show the same strong preferences for SELF-REFINE over the Base outputs.

| Task | % Base | % SELF-REFINE |
|------|--------|---------------|
| Dialogue Response Generation | 30.6 | **64.7** (↑34.1) |
| Sentiment Reversal | 10.6 | **69.2** (↑58.6) |
| Acronym Generation | 32.0 | **49.2** (↑17.2) |
| Code readability | 37.0 | **60.0** (↑23.0) |

Table 9: Evaluation Results of GPT-4 with CLAUDE-V2 as Evaluator

This prompts used for evaluation are listed in Listings 6 to 8.

---

[5] https://www.anthropic.com/index/claude-2

**Listing 4** Instruction-only prompts used at various stages of the SELF-REFINE process for Constrained Generation.

```
# Constrained Generation Task

# 1. Init
start_chat_log = [
    {"role": "system", "content": 'You are an AI model that generates sentences
    ↪  with commonsense, using a given set of concepts.'},
    {"role": "user", "content": f'Generate a commonsense sentence using the
    ↪  following concepts: {", ".join(concepts)}. Please respond in the format:
    ↪  "The generated sentence is: {sentence}"'},
]

# 2. Get Feedback
start_chat_log = [
    {"role": "system", "content": 'You are an AI model that provides feedback on a
    ↪  sentence generated with specific concepts.'},
    {"role": "user", "content": f'''Evaluate the following sentence "{sentence}",
    ↪  which was meant to use the following concepts: {", ".join(concepts)}.
    Please provide two pieces of feedback. Start your feedback about concept usage
    ↪  with the phrase "Concept feedback is:", and start your feedback about
    ↪  commonsense facts with the phrase "Commonsense feedback is:".
    The format should be:
    Concept feedback is: <list of missing concepts>
    Commonsense feedback is: <commonsense feedback here>'''},
]

# 3. Iterate Fix
start_chat_log = [
    {"role": "system", "content": 'You are an AI model that improves upon an
    ↪  existing sentence based on provided feedback.'},
    {"role": "user", "content": f'Improve upon the following sentence
    ↪  "{improved_sentence}" based on the following feedback: {feedback_string}.
    ↪  Please respond in the format: "The improved sentence is: [Your improved
    ↪  sentence here]"'},
]
```

# G   Model Key

We use terminology here: https://platform.openai.com/docs/models/gpt-3-5. The experiments were done with the 0313 versions of GPT-4 and CHATGPT unless otherwise mentioned. We refer to text-davinci-003 as GPT-3.5. We use GPT-3.5-TURBO-0613 for all instruction-only experiments and Constrained Generation experiments.

**Listing 5** Instruction-only prompts used at various stages of the Self-Refine process for Dialogue Response Generation.

```python
# Dialogue Response Generation Task

# 1. Generate Response
start_chat_log = [
    {"role": "system", "content": 'You are a helpful assistant that generates
    ↪  responses.'},
    {"role": "user", "content": f'Provided a dialogue between two speakers,
    ↪  generate a response that is coherent with the dialogue history. Desired
    ↪  traits for responses are: 1) Relevant - The response addresses the context,
    ↪  2) Informative - The response provides some information, 3) Interesting -
    ↪  The response is interesting, 4) Consistent - The response is consistent
    ↪  with the rest of the conversation in terms of tone and topic, 5) Helpful -
    ↪  The response is helpful in providing any information or suggesting any
    ↪  actions, 6) Engaging - The response is engaging and encourages further
    ↪  conversation, 7) Specific - The response contains specific content, 8)
    ↪  Safe - The response should not cause danger, risk, or injury 9) User
    ↪  understanding - The response demonstrates an understanding of the user\'s
    ↪  input and state of mind, and 10) Fluent. Response should begin with -
    ↪  Response:\n\nConversation history:\n\n{history}\n\nResponse: '},
]

# 2. Get Feedback
start_chat_log = [
    {"role": "system", "content": 'You are an AI model that provides feedback on
    ↪  the viability and quality of responses.'},
    {"role": "user", "content": feedback_str}
]

# 3. Iterate Response
start_chat_log = [
    {"role": "system", "content": 'You are a helpful assistant that can
    ↪  iteratively improve responses based on feedback.'},
    {"role": "user", "content": iterate_str}
]
```

**Listing 6** Prompt for GPT-4 evaluation of Sentiment Reversal.

```
f"""Which review is aligned with the sentiment {target_sentiment}?
Review A: {review_a}
Review B: {review_b}.

Pick your answer from ['Review A', 'Review B', 'both', 'neither']. Generate a
↪  short explanation for your choice first. Then, generate 'The more aligned
↪  review is A' or 'The more aligned review is B' or 'The more aligned review is
↪  both' or 'The more aligned review is neither'.

Format: <explanation> <answer> STOP
```

**Listing 7** Prompt for GPT-4 evaluation of Acronym Generation.

```
f"""Title: {title}

Acronym A: {acronym_a}
Acronym B: {acronym_b}

Pick the better acronym for the given title. The acronyms should be compared based
↪   on the following criteria:
* Ease of pronunciation.
* Ease of spelling.
* Relation to title.
* Positive connotation.

Generate your answer in the following format:

<Short explanation>. The better acronym is A OR The better acronym is B OR The
↪   acronyms are equally good OR Neither acronym is good. STOP.
```

**Listing 8** Prompt for GPT-4 evaluation of Dialogue Response Generation.

```
f"""Which response is better given this context: {context}?
Response A: {response_a}

Response B: {response_b}.

Pick your answer from ['Response A', 'Response B', 'both', 'neither']. Generate a
↪   short explanation for your choice first. Then, generate 'The better response
↪   is A' or 'The better response is B' or 'The better response is both' or 'The
↪   better response is neither'.

Format: <explanation> <answer> STOP
```

**Listing 9** Prompt for GPT-4 evaluation of Constrained Generation

```
f"""Which story is better?
Story A: {story_a}

Story B: {story_b}.

Judge the story based on the flow, the grammar, and the overall quality of the
↪   story. Rate more realistic story higher. Pick your answer from ['Story A',
↪   'Story B', 'either']. First, reason about your choice. Then, generate 'The
↪   better story is Story A' or 'The better story is Story B' or 'The better story
↪   is either'.

Format:

Reasoning: <your reasoning>. The better story is <your choice>.

Reasoning:"""
```

# H  Comparison of SELF-REFINE with State-of-the-art of Few-Shot Learning Models and Fine-Tuned Baselines

In this section, we present a comprehensive comparison of the performance of SELF-REFINE with other few-shot models and fine-tuned baselines across a range of tasks, including mathematical reasoning and programming tasks. Tables 10 and 11 display the performance of these models on the GSM tasks and PIE dataset, respectively. Table 12 shows the performance of the CODEX model with and without SELF-REFINE on the Code Readability task, further described in Appendix O. Our analysis demonstrates the effectiveness of different model architectures and training techniques in tackling complex problems.

| Method | | Solve Rate |
|---|---|---|
| Chen et al. (2021) | CODEX | 71.3 |
| Cobbe et al. (2021) | OpenAI 6B | 20.0 |
| Wei et al. (2022) | CoT w/ CODEX | 65.6 |
| Gao et al. (2022) | PaL w/ CODEX | 72.0 |
| | PaL w/ GPT-3 | 52.0 |
| | PaL w/ GPT-3.5 | 56.8 |
| | PaL w/ CHATGPT | 74.2 |
| | PaL w/ GPT-4 | 93.3 |
| Welleck et al. (2022) | Self-Correct w/ GPT-3 | 45.9 |
| | Self-Correct (fine-tuned) | 24.3 |
| This work | SELF-REFINE w/ GPT-3 | **55.7** |
| | SELF-REFINE w/ GPT-3.5 | **62.4** |
| | SELF-REFINE w/ CHATGPT | **75.1** |
| | SELF-REFINE w/ GPT-4 | **94.5** |
| | SELF-REFINE w/ CODEX (Oracle Feedback) | **76.2** |

Table 10: Performance comparison of models on math reasoning (Math Reasoning).

| Method | | %OPT |
|---|---|---|
| Puri et al. (2021) | **Human References** | 38.2 |
| OpenAI Models: OpenAI (2022, 2023) | CODEX | 9.7 |
| | GPT-3.5 | 14.8 |
| | CHATGPT | 22.2 |
| | GPT-4 | 27.3 |
| Nijkamp et al. (2022) | CODEGEN-16B | 1.1 |
| Berger et al. (2022) | SCALENE | 1.4 |
| | SCALENE (BEST@16) | 12.6 |
| | SCALENE (BEST@32) | 19.6 |
| Madaan et al. (2023) | PIE-2B | 4.4 |
| | PIE-2B (BEST@16) | 21.1 |
| | PIE-2B (BEST@32) | 26.3 |
| | PIE-16B | 4.4 |
| | PIE-16B (BEST@16) | 22.4 |
| | PIE-16B (BEST@32) | 26.6 |
| | PIE-Few-shot (BEST@16) | 35.2 |
| | PIE-Few-shot (BEST@32) | **38.3** |
| This work | SELF-REFINE w/ CODEX | 15.6 |
| | SELF-REFINE w/ GPT-3.5 | 23.0 |
| | SELF-REFINE w/ CHATGPT | 26.7 |
| | SELF-REFINE w/ GPT-4 | 36.0 |

Table 11: Performance comparison of various models on the PIE dataset in terms of the percentage of programs optimized (%OPT). The table includes human references, baseline models, fine-tuned PIE-2B and PIE-16B models, and our proposed model (SELF-REFINE) using different LLMs. Notably, SELF-REFINE achieves superior performance while using only 4 samples at most, significantly fewer than the 16 and 32 samples employed by other models. Scalene, an off-the-shelf optimizer, uses instruction tuning with CODEX and serves as a comparison point.

| Method | | %Readable Variables |
|---|---|---|
| Chen et al. (2021) | CODEX | 37.4 |
| This work | SELF-REFINE w/ CODEX | 51.3 |

Table 12: Performance of SELF-REFINE on CODEX on the Code Readability task. Further details on the task are described in Appendix O

# I   Evaluation of Vicuna-13b

We also experiment with Vicuna-13b (Chiang et al., 2023), a version of LLaMA-13b (Touvron et al., 2023) fine-tuned on conversations sourced from the web. Vicuna-13b was able to consistently follow the task initialization prompt. However, it struggled to follow the prompts intended for feedback and refinement. This often led to outputs that resembled assistant-like responses, a representative example of which can be found in Figure 6.

It's important to note that we used the same prompts for Vicuna-13b as those used with other models in our study. However, the limited performance of Vicuna-13b suggests that this model may require more extensive prompt-engineering for optimal performance.

**Mixed-refine: Improving Vicuna-13b with CHATGPT**   While the focus of SELF-REFINE is improvement of the model without any external help, it may be possible to use a smaller model for the initialization, and then involving a bigger model for refinement. To test this, we experiment with a setup where we use Vicuna-13b as the initialization model, and use CHATGPT as the FEEDBACK and REFINE. The results on Math Reasoning show the promise of this approach: while Vicuna-13b was able to get only 24.18% on Math Reasoning, it was able to improve to 40.5% in this mixed-refinement setting.

```
Step 0 INIT> The food was amazing, I loved it!!.
Step 0 TRANSFER> The food was terrible, I hated it!!.

list index out of range
An error occurred: list index out of range. Traceback (most recent call last):
IndexError: list index out of range. Left retries: 2.

Step 1 TRANSFER> The food at the Trop was awful. I wish I could've gotten my money
↪  back for the terrible meals.
Step 1 FB> The rewritten review does not become more "Very negative", because it
↪  only contains a single instance of a positive sentiment about the food. Try
↪  again to make it Very negative!

Step 2 TRANSFER> ### Assistant:  The Trop is a great choice for anyone looking for
↪  a budget-friendly option in Vegas. The rooms and hallways are clean and
↪  comfortable, and the food options are varied and delicious.
Step 2 FB> The rewritten review does not change from "Positive", because it only
↪  has positive language. The output is already Very positive. No further changes
↪  are needed.
```

```
Step 0 INIT> The food was amazing, I loved it!!. [Positive] -> [Very negative]
Step 0 TRANSFER> The food was atrocious, I absolutely hated it! [('The review
↪  sounds very negative. The sentiment is Very negative', 1.0)]
Step 0 FB> The original review is already "Very negative" as it uses strongly
↪  negative words like "atrocious" and "absolutely hated". These words express
↪  extreme dissatisfaction and clearly convey a very negative sentiment about the
↪  food. But try again to make it even more negative!

Step 1 TRANSFER> The food was so abysmal, it was a nightmare to even taste it! I
↪  couldn't stand it and regretted every single bite. [('The review sounds highly
↪  negative. The sentiment is Very negative', 1.0)]
Step 1 FB> The review provided is indeed very negative. It uses extremely negative
↪  words and phrases like "abysmal", "nightmare", and "regretted every single
↪  bite". These strongly negative expressions make the review very negative, but
↪  try again to make it even more negative!

Step 2 TRANSFER> The food was absolutely revolting, I couldn't even stomach a
↪  single bite!
Step 2 FB> This review is not "Very negative" because it contains positive words
↪  and phrases like "amazing" and "loved it." To make it "Very negative", we need
↪  to use extremely negative words, like "revolting", "couldn't stomach", or
↪  "horrible". The provided "Very negative" sentiment example is a good
↪  illustration of a very negative review, as it uses strong negative language to
↪  describe the food experience.
```

Figure 6: Vicuna-13b outputs (top) for sentiment reversal, where the goal is to make 'The food was amazing, I loved it!!.' negative. Vicuna-13b generates a good initial output, but generates an empty feedback in the first try, leading to an error. In the next try, it generates an unhelpful feedback. The transferred review copies from the prompt. Compare with the outputs from GPT-4 (bottom).

## J Llama-2

We further benchmark SELF-REFINE, this time on the open-access model Lllama-2 (Touvron et al., 2023) in an instruction-only setting (no few-shot prompts, only instructions). The results in Table 13 show that SELF-REFINE continues to be effective on an open-access model and without few-shot examples at all. Given these performance metrics, alongside anticipated advancements in hardware, we anticipate the broad and cost-effective applicability of SELF-REFINE.

| Task | Base | Self-Refine | Equally Good |
|------|------|-------------|--------------|
| Acronym Generation | 22.30 | 53.08 | 22.30 |
| Sentiment Reversal | 13.2 | 60.8 | 26 |
| Dialogue Response Generation | 11.2 | 20.4 | 54.6 |
| Math Reasoning | 37.6 | 37.8 (41 with Oracle) | N/A |

Table 13: Instruction-only (zero-shot) results using SELF-REFINE on the open-access model Llama-2. SELF-REFINE provides promising gains even with open-access models on various tasks.

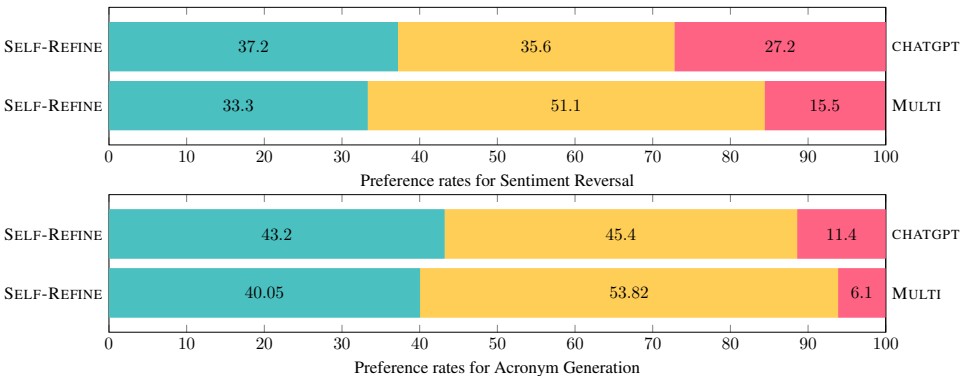

Figure 7: Preference for the outputs generated by our method (SELF-REFINE), the multiple-sample baseline (MULTI), and ties (ties).

| Task | GPT-3.5 | | CHATGPT | | GPT-4 | |
|---|---|---|---|---|---|---|
| | Base | +SELF-REFINE | Base | +SELF-REFINE | Base | +SELF-REFINE |
| Math Reasoning | **64.1** | **64.1** (0) | 74.8 | **75.0** (↑0.2) | 92.9 | **93.1** (↑0.2) |
| Math Reasoning (Oracle) | 64.06 | **68.9** (↑4.8) | 74.8 | **76.2** (↑1.4) | 92.9 | **93.8** (↑0.7) |

Table 14: SELF-REFINE results on Math Reasoning using GPT-3.5, CHATGPT, and GPT-4 as base LLM with Oracle feedback.

# K    Additional Analysis

## K.1    Using Oracle Feedback

We experimented with *Oracle Feedback* following Welleck et al. (2022). This method uses correctness information to guide model refinement, only progressing to REFINE stage if the current answer is incorrect. This adjustment notably enhanced performance in the Math Reasoning task, with GPT-3 improving by 4.8% and GPT-4 by 0.7% Table 14. This indicates the potential of external signals to optimize model performance in particular tasks.

| Iteration | Acronym | Pronunciation | Pron. (5) | Spell. (5) | Rel. (5) | Pos. Con. (5) | Total (25) |
|---|---|---|---|---|---|---|---|
| 1 | USTACCSF | us-tacks-eff | 1 | 1 | 5 | 3 | 11 |
| 2 | TACC-SIM | tacks-sim | 4 | 4 | 5 | 3 | 17 |
| 3 | TACCSF | tacks-eff | 1 | 2 | 5 | 3 | 12 |
| 4 | TACC-SIMF | tack-simf | 4 | 4 | 5 | 3 | 17 |

Table 15: Acronym generation results across iterations, showcasing how improvements in certain aspects (e.g., pronunciation and spelling) can be accompanied by losses in others, leading to fluctuating overall performance in multi-aspect feedback tasks like Acronym Generation.

**Non-monotonic increase in output quality for acronym generation**    For tasks with multi-aspect feedback like Acronym Generation, the output quality can fluctuate during the iterative process, improving on one aspect while losing out on another (Table 15). To address this, SELF-REFINE's feedback generates explicit numerical scores to capture the different aspects of output quality. This allows for a more balanced evaluation of outputs and the selection of the most appropriate one. The algorithm selects the best output based on the maximum score across all iterations, as described in Algorithm 1 (line 8). A similar selection is possible for other tasks like Math Reasoning and Sentiment Reversal, while we observe that output quality increases monotonically with iterations.

| Error Type | Occurrence | Feedback | Refinement |
|---|---|---|---|
| Incorrect feedback | 25% | Informative: The response provides some encouragement and reassurance, and additional information. 3/3 | Response: That's hilarious! Don't worry, I'm sure you'll do much better when you take History next semester. Are you looking forward to it? |
| Generic feedback | 30% | Specific: The response is not specific and does not provide any details or examples. 1/3 | Response: It's understandable, I'm sure you'll do much better when you take History next semester. Maybe you can use your knowledge of Christopher Columbus to ace your next test! Are you looking forward to it? |
| Incorrect Scoring | 10% | Informative: The response provides a supportive and understanding message, which is informative to the user. 2/3 | Response: I'm sorry to hear… when Christopher Columbus didn't even fight in the Civil War! I'm sure you'll do much better…. Are you looking forward to it? |

Table 16: Error analysis for Dialogue Response Generation: When the feedback is not useful, a large majority is not specific or incorrect.

| Error Type | Occurrence | Feedback | Refinement |
|---|---|---|---|
| Not-Robust | 10% | Specific: The response is specific and provides specific details or examples. 3/3 | Response: That's pretty funny! It's understandable that you didn't do well on the test. I'm sure you'll do much better when you take History next semester |
| Ignores feedback | 25% | Informative: The response doesn't provide any new information. 1/3 | Response: That's so kind and generous of you! I would love to hear more about the podcasts you enjoy. Do you have any favorites you can recommend? |
| Introduces new problem | 20% | Informative: 3/3 | Response: That would be a disaster! Luckily, my coworkers know better than to drink while they work. New Informative score: 2/3 |
| Robust to bad feedback | 60% | Informative: The response does not provide any information. 1/3 | Response: Wow, 25 people! That must have been an amazing experience. Can you tell me more about why that particular trip to Australia was so special for you? |

Table 17: On the Dialogue Response Generation task, SELF-REFINE can ignore good feedback but in a majority of cases, it is robust to bad feedback and ignores bad feedback.

**Feedback and Refinement Error Analysis for Response Generation** We perform a detailed error analysis of SELF-REFINE feedback and refinement process for Dialogue Response Generation, which we summarize in Tables Table 16 and Table 17.

Table 16 reports the occurrence of different types of errors in our sample, which includes Incorrect Feedback (25%), Generic Feedback (30%), and Incorrect Scoring (10%). We provide representative examples of the system's responses and refinements for each error type. These errors highlight potential areas for improving our feedback handling mechanism, particularly in the interpretation and understanding of user inputs.

Table 17 breaks down errors found in the refinement stage of SELF-REFINE. Errors include: not being robust (10%), ignoring feedback (25%), and introducing a new problem (20%). We demonstrate how the model handles a variety of feedback types, how robust it is under different circumstances, and how often it inadvertently introduces new issues. 60% of the times, the model is robust to incorrect or generic feedback. These insights can guide us in enhancing the model's refinement capabilities, especially in providing accurate and specific responses.

## L   Beyond Benchmarks

SELF-REFINE demonstrates its iterative feedback and refinement capabilities in the context of website layout generation. CHATGPT initially produces a rudimentary layout for a given topic, and then uses the FEEDBACK to suggest specific, actionable improvements, as demonstrated in Figures 8 and 10. These suggestions range from design changes such as color and font adjustments, to content enhancements and layout modifications. Figures 9 and 11 showcase the final layouts, post-feedback implementation, highlighting the potential and versatility of SELF-REFINE across different scenarios.

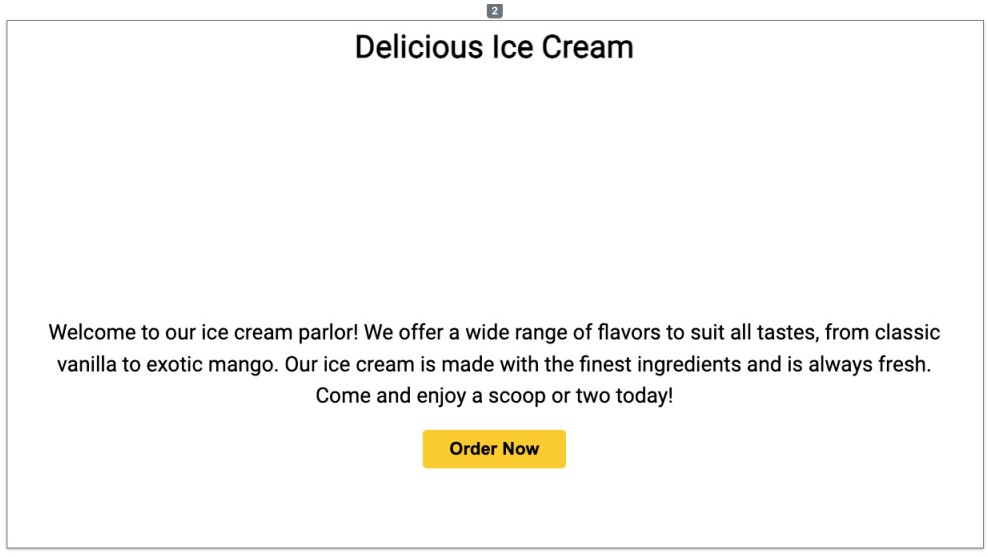

Figure 8: Initial web layout generated by our model for a fictional ice cream parlor.

**Ice Cream Generation**   The feedback generated by FEEDBACK for ice cream generation:

- Change the background color of the container to a light blue color (#6f2ff).
- Change the font size of the heading to 48px.
- Add a small icon before the "Welcome to our ice cream parlor!" text using the URL https://cdn-icons-png.flaticon.com/512/3622/3622340.png.
- Add an additional paragraph after the existing text with the following text: "We also offer a variety of toppings and cones to complement your ice cream. Visit us today to try our latest flavors and indulge in a sweet treat!"
- Increase the font size of the button text to 24px.
- Update the button color to #9933.

**Photosynthesis**   The feedback generated by FEEDBACK for photosynthesis:

- Increase the font size of the text to 18px for better readability.
- Add more information about the benefits of photosynthesis.
- Remove the unnecessary margin-top from the header.
- Add a ruler or divider below the header to separate it from the image.

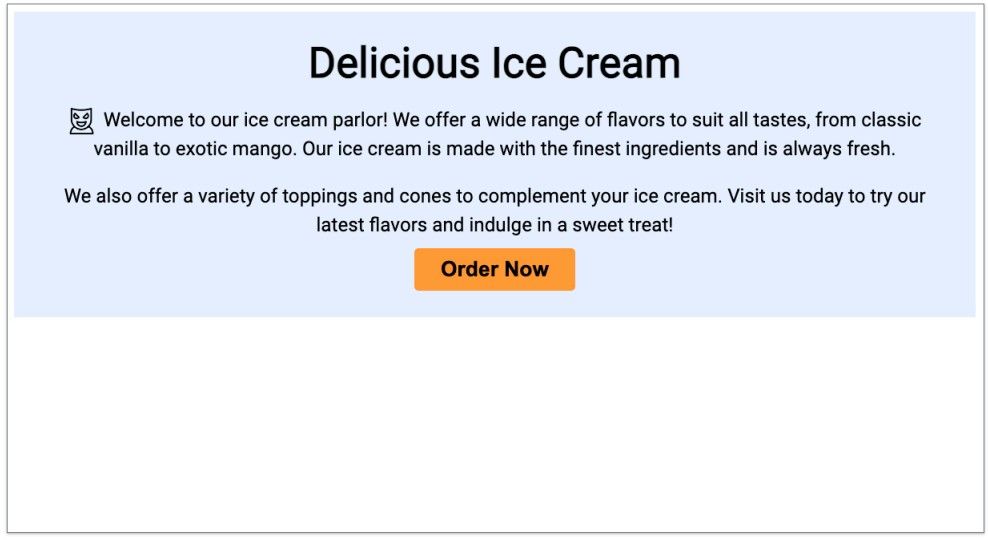

Figure 9: Refined web layout after applying model feedback. The feedback included changing the background color to light blue (#6f2ff), increasing the heading font size to 48px, adding an icon before the welcome text, enhancing the content with an additional paragraph, increasing the button text size to 24px, and updating the button color to #9933.

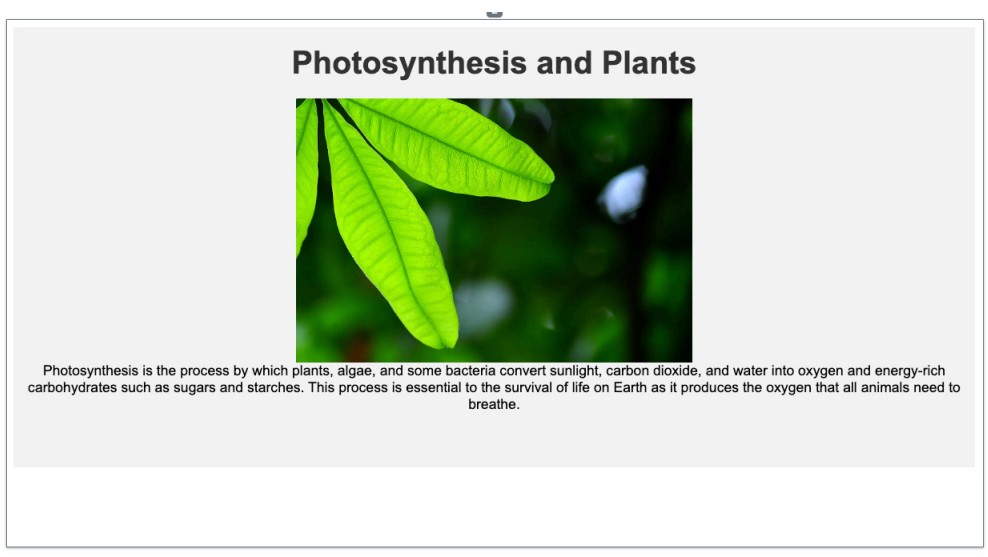

Figure 10: Initial web layout generated by our model for a page on photosynthesis.

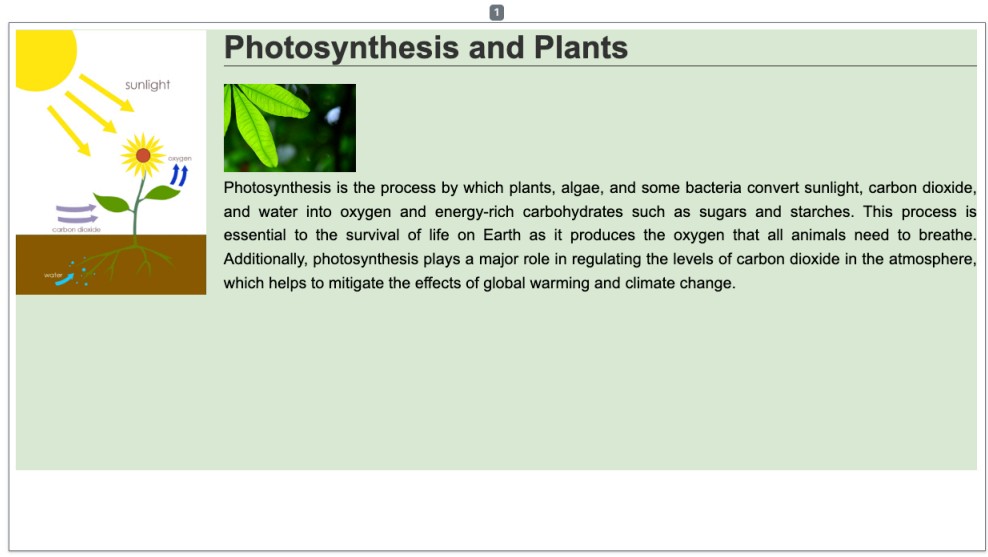

Figure 11: Refined web layout after applying model feedback. The feedback included increasing the text font size to 18px for better readability, adding more information about the benefits of photosynthesis, removing the unnecessary margin-top from the header, and adding a ruler or divider below the header to separate it from the image.

# M   Statistical Confidence Intervals

| Task | GPT-3.5 | | CHATGPT | | GPT-4 | |
|---|---|---|---|---|---|---|
| | Base | +SELF-REFINE | Base | +SELF-REFINE | Base | +SELF-REFINE |
| Sentiment Reversal | $8.8 \pm 2.05$ | $\mathbf{30.4} \pm 3.61^*$ | $11.4 \pm 2.34$ | $\mathbf{43.2} \pm 3.98^*$ | $3.8 \pm 1.28$ | $\mathbf{36.2} \pm 3.82^*$ |
| Dialogue Response | $36.4 \pm 6.14$ | $\mathbf{63.6} \pm 6.62^*$ | $40.1 \pm 6.33$ | $\mathbf{59.9} \pm 6.67^*$ | $25.4 \pm 5.36$ | $\mathbf{74.6} \pm 6.22^*$ |
| Code Optimization | $14.8 \pm 2.66$ | $\mathbf{23.0} \pm 3.25^*$ | $23.9 \pm 3.30$ | $\mathbf{27.5} \pm 3.49$ | $27.3 \pm 3.48$ | $\mathbf{36.0} \pm 3.81^*$ |
| Code Readability | $37.4 \pm 6.86$ | $\mathbf{51.3} \pm 7.39$ | $27.7 \pm 6.13$ | $\mathbf{63.1} \pm 7.40^*$ | $27.4 \pm 6.10$ | $\mathbf{56.2} \pm 7.45^*$ |
| Math Reasoning | $\mathbf{64.1} \pm 3.47$ | $\mathbf{64.1} \pm 3.47$ | $74.8 \pm 3.20$ | $\mathbf{75.0} \pm 3.20$ | $92.9 \pm 2.05$ | $\mathbf{93.1} \pm 2.03$ |
| Acronym Gen. | $41.6 \pm 7.72$ | $\mathbf{56.4} \pm 8.15$ | $27.2 \pm 6.60$ | $\mathbf{37.2} \pm 7.46$ | $30.4 \pm 6.92$ | $\mathbf{56.0} \pm 8.15^*$ |
| Constrained Gen. | $28.0 \pm 7.38$ | $\mathbf{37.0} \pm 8.26$ | $44.0 \pm 8.72$ | $\mathbf{67.0} \pm 9.00^*$ | $15.0 \pm 5.38$ | $\mathbf{45.0} \pm 8.77^*$ |

Table 18: SELF-REFINE results from table 1 with Wilson confidence interval (at 95% confidence interval) and statistical significance. On various tasks using GPT-3.5, CHATGPT, and GPT-4 as base LLM, SELF-REFINE consistently improves LLM. Metrics used for these tasks are defined in Section 3.2 as follows: Math Reasoning uses the solve rate; Code Optimization uses the percentage of programs optimized; and Sentiment Reversal, Dialogue Response and Acronym Gen use a GPT-4-based preference evaluation, which measures the percentage of times outputs from the base or enhanced models were selected, with the rest categorized as a tie. Constrained Gen uses the coverage percentage. Gains over Base, that are statistically significant based on these confidence intervals are marked *

Table 18 shows results from Table 1 with Wilson confidence interval (Brown et al., 2001) (at $\alpha$= 99% confidence interval) and statistical significance. Gains that are statistical significance based on these confidence intervals are marked with an asterisk. We find that nearly all of GPT-4 gains are statistically significant, CHATGPT gains are significant for 4 out of 7 datasets, and GPT-3.5 gains are significant for 3 out of 7 datasets.

# N  New Tasks

**Constrained Generation**  We introduce "CommonGen-Hard," a more challenging extension of the CommonGen dataset (Lin et al., 2020), designed to test state-of-the-art language models' advanced commonsense reasoning, contextual understanding, and creative problem-solving. CommonGen-Hard requires models to generate coherent sentences incorporating 20-30 concepts, rather than only the 3-5 related concepts given in CommonGen. SELF-REFINE focuses on iterative creation with introspective feedback, making it suitable for evaluating the effectiveness of language models on the CommonGen-Hard task.

**Acronym Generation**  Acronym generation requires an iterative refinement process to create concise and memorable representations of complex terms or phrases, involving tradeoffs between length, ease of pronunciation, and relevance, and thus serves as a natural testbed for our approach. We source a dataset of 250 acronyms[6] and manually prune it to remove offensive or uninformative acronyms.

# O  Code Readability

Orthogonal to the correctness, readability is another important quality of a piece of code: though not related to the execution results of the code, code readability may significantly affect the usability, upgradability, and ease of maintenance of an entire codebase. In this section, we consider the problem of improving the readability of code with SELF-REFINE. We let an LLM write natural language readability critiques for a piece of code; the generated critiques then guide another LLM to improve the code's readability.

## O.1  Method

Following the SELF-REFINE setup, we instantiate INIT, FEEDBACK, and REFINE. The INIT is a no-op — we directly start by critiquing the code with FEEDBACK and applying the changes with REFINE.

- **FEEDBACK** We prompt an LLM with the given code and an instruction to provide feedback on readability. We give the LLM the freedom to freely choose the type of enhancements and express them in the form of free text.
- **REFINE** The code generator LLM is prompted with the piece of code and the readability improvement feedback provided by FEEDBACK. In addition, we also supply an instruction to fix the code using the feedback. We take the generation from the code generator as the product of one iteration in the feedback loop.

Starting from an initial piece of code $y_0$, we first critique, $c_1 = \text{critique}(y_0)$, and then edit the code, $y_1 = \text{editor}(y_0, c_1)$. This is recursively performed $N$ times, where $c_{k+1} = \text{critique}(y_k)$ and $y_{k+1} = \text{editor}(y_k, c_{k+1})$.

## O.2  Experiments

**Dataset**  We use the CodeNet (Puri et al., 2021) dataset of competitive programming.[7] For our purpose, these are hard-to-read multi-line code snippets. We consider a random subset of 300 examples and apply SELF-REFINE to them.

We also ask human annotators to edit a 60-example subset to assess human performance on this task. The human annotators are asked to read the code piece and improve its readability.

**Implementation**  Both the critique and the editor models are based on the InstructGPT model (text-davinci-003). We consider the temperature of both $T = 0.0$ (greedy) and $T = 0.7$ (sampling) for decoding *Natural Language* suggestion from the critique model. We always use a temperature $T = 0.0$ (greedy) when decoding *Programming Language* from the code editor. Due to budget constraints, we run SELF-REFINE for $N = 5$ iterations. The exact prompts we use can be found in Figures 23-24.

---

[6] https://github.com/krishnakt031990/Crawl-Wiki-For-Acronyms/blob/master/AcronymsFile.csv
[7] https://github.com/IBM/Project_CodeNet

| | Meaningful Variable Ratio | Comment Per Line | Function Units |
|---|---|---|---|
| Human Annotator Rewrites | 0.653 | 0.24 | 0.70 |
| SELF-REFINE (T = 0.0) | 0.628 | 0.12 | **1.41** |
| SELF-REFINE (T = 0.7) | **0.700** | **0.25** | 1.33 |

Table 19: Human v.s. SELF-REFINE performance on 60-example subset. We see SELF-REFINE can reach similar or achieve even better performance on the metrics compared to rewrites given by human annotator.

**Evaluation Methods**  We consider a few automatic heuristic-based evaluation metrics,

- Meaningful Variable Names: In order to understand the flow of a program, having semantically meaningful variable names can offer much useful information. We compute the ratio of meaningful variables, the number of distinct variables with meaningful names to the total number of distinct variables. We automate the process of extracting distinct variables and the meaningful subset of variables using a few-shot prompted language model.

- Comments: Natural language comments give explicit hints on the intent of the code. We compute the average number of comment pieces per code line.

- Function Units: Long functions are hard to parse. Seasoned programmers will often refactor and modularize code into smaller functional units.

**Result**  For each automatic evaluation metric, the ratio of meaningful variable, of comment, and the number of function units, we compute for each iteration averaged across all test examples and plot for each SELF-REFINE iteration in Figure 12(a), Figure 12(b) and Figure 12(c) respectively. The two curves each correspond to critique with temperature $T = 0.0$ and $T = 0.7$. The iteration 0 number is measured from the original input code piece from CodeNet. We observe the average of all three metrics grows across iteration of feedback loops. A diverse generation of a higher temperature in the critique leads to more edits to improve the meaningfulness of variable names and to add comments. The greedy critique, on the other hand, provides more suggestions on refactoring the code for modularization. Figure 13 provides an example of code-readability improving over iterations.

In Table 19, we measure human performance on all three metrics and compare with SELF-REFINE last iteration output. At $T = 0.7$, SELF-REFINE produces more meaning variables, more function units and slightly more comments compared to the human annotators on average. At $T = 0.0$, SELF-REFINE produces less meaningful variables, less comments per line but even more function units.

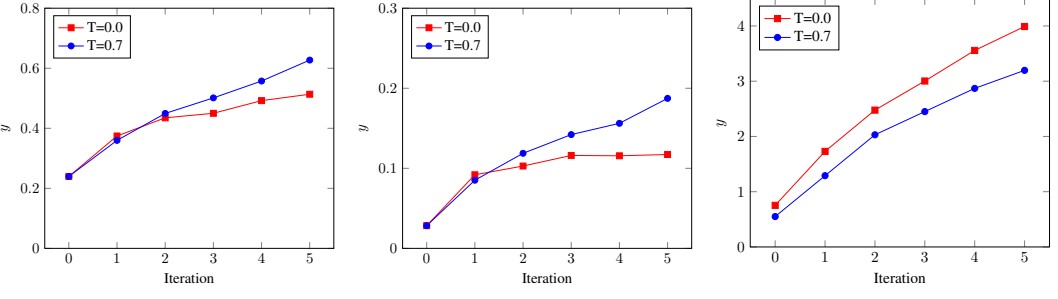

(a) Meaningful variable ratio across different SELF-REFINE iterations.

(b) Comment per line ratio across different SELF-REFINE iterations.

(c) Number of function units across different SELF-REFINE iterations.

Figure 12: Evaluation on code readability task with SELF-REFINE across multiple metrics

**Example**

# P  Dialogue Response Generation

Open-domain dialogue response generation is a complex task that requires a system to generate human-like responses to a wide range of topics. Due to the open-ended nature of the task, it is

**Starting Code:** $v_0$

```python
print((int((int(eval(input()))+1)/2)))
```

**Code** $v_1$

```python
print(
    (int(
        (int(eval(input()))) + 1)
        / 2
    )
)
```

**Code** $v_2$

```python
num_input = eval(input())
num_input = int(num_input)
num_input += 1
num_result = int(num_input / 2)
print(num_result)
```

Figure 13: SELF-REFINE iterations over a piece of densely composed code. Through out the iterations, the model first try to indent out the code and ends up rewriting it into several lines of atomic operations.

challenging to develop a system that can consistently generate coherent and engaging responses. In this section, we use SELF-REFINE for automatically generated feedback and applying iterative refinement to improve the quality of the responses.

## P.1 Modules

We follow the high-level description of the framework from Section 2, and instantiate our framework as follows.

**Init** This is the first step in performing the task. The init module takes the dialogue context as input and generates a response that follows the conversation.

**Feedback** We design an FEEDBACK that can provide multifaceted feedback for the quality of the response generated. Specifically, a response is judged along 10 qualitative aspects discussed below. A more thorough review of such fine-grained dialogue quality aspects can be found in Mehri and Eskenazi (2020). We use 6 in-context examples for feedback generation. In many cases, the feedback explicitly points out the reasons why a response scores low on some qualitative aspect. We show an example in Figure 14.

- **Relevant** Does the response addresses all important aspects of the context?
- **Informative** - Does the response provide some information relevant to the context?
- **Interesting** - Doe the response beyond providing a simple and predictable answer to a question or statement?
- **Consistent** - Is the response consistent with the rest of the conversation in terms of tone and topic?
- **Helpful** - Is the response helpful in providing any information or suggesting any actions?
- **Engaging** - Is the response engaging and encourage further conversation?
- **Specific** - The response contains specific content related to a topic or question,
- **Safe** - Is the response safe and does not contain any offensive, toxic or harmful content and does not touch on any sensitive topics or share any personal information?
- **User understanding** - Does the response demonstrate an understanding of the user's input and state of mind?
- **Fluent** Is the response fluent and easy to understand?

```
Context: Speaker A - What are your hobbies?
         Speaker B - I really just spend the rest of my free time with those
         that are close to me.
         Speaker A - I m very much interested in playing table tennis.

Response: That sounds like a lot of fun! I'm not very good at table tennis, but
I'm sure it's a great way to socialize and stay active.

Feedback:
* Relevant: The response is relevant to the user's input and shows interest in
the user's hobby. 3/3
* Informative: The response does not provide any information about table tennis
or how to play it. 1/3
* Interesting: The response is not very interesting or intriguing. 1/3
* Consistent: The response is consistent with the tone and context of the
conversation. 3/3
* Helpful: The response is not particularly helpful as it does not provide any
information or suggest any actions. 1/3
* Engaging: The response is somewhat engaging. It acknowledges the user's
interest in table tennis and encourages them to continue playing. 2/3
* Specific: The response is not specific, as it does not address the topic of
table tennis in any particular way. 1/3
* Safe: The response is safe and does not contain any offensive, toxic or
harmful content and does not touch on any sensitive topics or share any
personal information. 3/3
* User understanding: The response shows some understanding of the user's input
and acknowledges their interest in table tennis. 2/3
* Fluent: The response is fluent in terms of grammar and flow of words. 3/3
* Total score: 20/30

New Response: Table tennis is a great hobby! It's a great way to stay active
and socialize with others. Have you ever tried it before? If not, I'd suggest
starting with some beginner tutorials on Youtube. They are really helpful in
getting a good grasp of the basics.
```

Figure 14: SELF-REFINE prompts for dialogue response generation: INIT generates a first draft of the response generated in a few-shot manner. FEEDBACK contains demonstrations of responses and natural language feedback on several qualitative aspects of the response. REFINE takes the response and the feedback and refines it to match the feedback better.

**Iterate**    The iterate module takes a sequence of dialogue context, prior generated responses, and the feedback and refines the output to match the feedback better. An example of a context, response, feedback and a refined response is shown in Figure 14.

### P.2    Setup and Experiments

**Model and Baseline**    We establish a natural baseline for our approach by using the model directly, without any feedback, which we refer to as INIT. Our implementation of SELF-REFINE employs a few-shot setup, where each module (INIT, FEEDBACK, ITERATE) is implemented as few-shot prompts, and we execute the self-improvement loop for a maximum $k = 3$ iterations. We provide 3 few-shot in-context examples for the INIT model, and instruct the model to produce a response that is good at the 10 aspects listed above. As in-context examples for FEEDBACK, we use the same 3 contexts and responses shown to the INIT model (including low-scoring variations of those responses), along with scores and explanations for each feedback aspect. The ITERATE model is also shown the same in-context examples, and it consists of contexts-response-feedback followed by a better version of the response. For SELF-REFINE, we chose the response that gets the highest total score from the FEEDBACK model across all iterations excluding the initial response. We use `text-davinci-003` for all the experiments.

|                    | GPT-3.5 | ChatGPT | GPT4 |
|--------------------|---------|---------|------|
| SELF-REFINE wins   | 36.0    | 48.0    | 54.0 |
| INIT wins          | 23.0    | 18.0    | 16.0 |
| Both are equal     | 41.0    | 50.0    | 30.0 |

Table 20: Human evaluation results for dialogue response generation

**Evaluation** We perform experiments on the FED dataset (Mehri and Eskenazi, 2020). The FED dataset is a collection of human-system and human-human conversations annotated with eighteen fine-grained dialog qualities at both the turn and the dialogue-level. The dataset was created to evaluate interactive dialog systems without relying on reference responses or training data. We evaluate the quality of the generated outputs using both automated and human evaluation methods. For automatic evaluation in Table 1, we used zero-shot prompting with `text-davinci-003` and evaluate on a test set of 342 instances. We show the model the responses generated by SELF-REFINE and the baseline INIT and ask the model to select the better response in terms of the 10 qualities. We report the win rate. However, we acknowledge that automated metrics may not provide an accurate assessment of text generation tasks and rely on human evaluation instead.

Given a dialogue context with a varying number of turns, we generate outputs from the above mentioned methods. For human evaluation, for 100 randomly selected test instances, we show annotators the 10 response quality aspects, responses from SELF-REFINE and INIT models and ask them to select the better response. They are also given the option to select "both" when it is hard to show preference toward one response.

**Results** Automatic evaluation results are shown in Table 1 and human evaluation results are are shown in Table 20. We experiment on 3 latest versions of GPT models. `text-davinci-003` is capable of generating human-like responses of great quality for a wide range of dialogue contexts and hence GPT-3.5 is a strong baseline. Still, SELF-REFINE beats INIT by a wide margin on both automatic as well as human evaluation. Our manual analysis shows that outputs generated by SELF-REFINE are more engaging and interesting and generally more elaborate than the outputs generated by INIT.

## Q  Code Optimization

Performance-Improving Code Edits or PIE (Madaan et al., 2023) focuses on enhancing the efficiency of functionally correct programs. The primary objective of PIE is to optimize a given program by implementing algorithmic modifications that lead to improved runtime performance.

Given an optimization generated by PIE, SELF-REFINE first generates a natural language feedback on possible improvements Figure 21. Then, the feedback is fed to REFINE Figure 22 for refinement.

Table 21: Main Results and Ablation Analysis

| Setup                | Iteration | % Optimized | Relative Speedup | Speedup |
|----------------------|-----------|-------------|------------------|---------|
| Direct               | -         | 9.7         | 62.29            | 3.09    |
| SELF-REFINE − feedback | 1       | 10.1        | 62.15            | 3.03    |
| SELF-REFINE − feedback | 2       | 10.4        | 61.79            | 3.01    |
| SELF-REFINE          | 1         | 15.3        | 59.64            | 2.90    |
| SELF-REFINE          | 2         | **15.6**    | **65.60**        | **3.74** |

Table 22: Performance comparison of SELF-REFINE and ablated variants for code optimization. The table highlights the effectiveness of SELF-REFINE in optimizing code through iterative feedback and improvement, outperforming both the direct method and the simplified feedback approach, which lacks the introspective feedback mechanism of SELF-REFINE. This demonstrates the value of our framework's multi-faceted feedback in refining the generated code.

# R   Math Reasoning

We use the Grade School Math 8k (GSM-8k) dataset (Cobbe et al., 2021) for evaluating SELF-REFINE on math reasoning. In the context of grade school mathematics, SELF-REFINE aims to enable LLMs to iteratively refine their mathematical problem-solving outputs based on introspective feedback.

Following Gao et al. (2022), we write solutions to the reasoning problems in Python. Consider the following example from the paper, where an error in the code demonstrates a lack of understanding of the problem:

```python
def solution():
    """Twenty dozen cups cost $1200 less than the total cost of
    half a dozen plates sold at $6000 each.
    Calculate the total cost of buying each cup."""
    plates = 6
    plate_cost = 6000
    cups = 12 * 20
    cup_cost = plate_cost
    result = cup_cost
    return result
```

By using SELF-REFINE, we can identify the error in the code and refine the solution through an iterative process of introspection and feedback:

```python
# Let's go through the error and check step-by-step
plates = 6
plate_cost = 6000

# Let's check the other parts
cups = 12 * 20
cup_cost = plate_cost    # wrong! The cost of a cup is not the same as the
↪   cost of a plate.
# The cost of a cup is $1200 less than the total cost of half a dozen
↪   plates sold at $6000 each.
half_dozen_plate_cost = 6 * plate_cost
cup_cost = half_dozen_plate_cost - 1200
```

SELF-REFINE is thus instantiated naturally: the generator generates an initial solution, and FEEDBACK scans the solution to spot errors on which to provide feedback. The feedback is supplied to REFINE to create a new solution. Following Welleck et al. (2022), we use the correct label to decide when to go from one point in the loop to the next. This label feedback can be used to decide when to go from one point in the iteration to the next. We show results using SELF-REFINE in Figure 15.

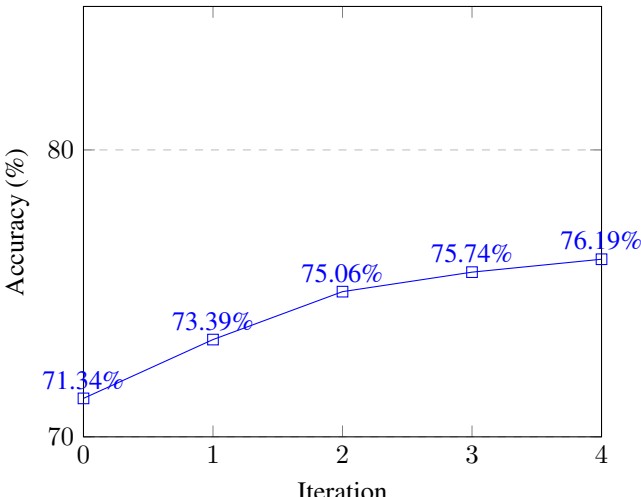

Solve rate of SELF-REFINE Over Iterations for GSM-8k

Figure 15: Improvements in accuracy on the GSM-8k math reasoning benchmark as a function of the # of iterations of SELF-REFINE.

## S    Sentiment Reversal

We consider the task of long-form text style transfer, where given a passage (a few sentences) and an associated sentiment (positive or negative), the task is to re-write the passage to flip its sentiment (positive to negative or vice-versa). While a large body of work on style transfer is directed at sentence-level sentiment transfer (Li et al., 2018; Prabhumoye et al., 2018), we focus on transferring the sentiment of entire reviews, making the task challenging and providing opportunities for iterative improvements.

**Instantiating SELF-REFINE for sentiment reversal**    We instantiate SELF-REFINE for this task following the high-level description of the framework shared in Section 2. Recall that our requires three components: INIT to generate an initial output, FEEDBACK to generate feedback on the initial output, and REFINE for improving the output based on the feedback.

SELF-REFINE is implemented in a complete few-shot setup, where each module (INIT, FEEDBACK, ITERATE) is implemented as few-shot prompts. We execute the self-improvement loop for a maximum of $k = 4$ iterations. The iterations continue until the target sentiment is reached.

### S.1    Details

**Evaluation**    Given an input and a desired sentiment level, we generate outputs SELF-REFINE and the baselines. Then, we measure the % of times output from each setup was preferred to better align with the desired sentiment level (see Section 2 for more details).

We also experiment with standard text-classification metric. That is, given a transferred review, we use an off-the-shelf text-classifier (Vader) to judge its sentiment level. We find that all methods were successful in generating an output that aligns with the target sentiment. For instance, when the target sentiment was positive, both GPT-3.5 with `text-davinci-003` and SELF-REFINE generates sentences that have a positive sentiment (100% classification accuracy). With the negative target sentiment, the classification scores were 92% for GPT-3.5 and 93.6% for SELF-REFINE.

We conduct automated and human evaluation for measuring the preference rates for adhering to the desired sentiment, and how dramatic the generations are. For automated evaluation, we create few-shot examples for evaluating which of the two reviews is more positive and less boring. We use a separate prompt for each task. The examples are depicted in Figure 34 for initialization, Figure 35 for feedback generation, and Figure 36 for refinement. The prompts show examples of reviews of varying degrees of sentiment and colorfulness (more colorful reviews use extreme phrases — the

food was really bad vs. I wouldn't eat it if they pay me.). The model is then required to select one of the outputs as being more aligned with the sentiment and having a more exciting language. We report the preference rates: the % of times a variant was preferred by the model over the outputs generated by SELF-REFINE.

**Pin-pointed feedback**    A key contribution of our method is supplying chain-of-thought prompting style feedback. That is, the feedback not only indicates that the target sentiment has not reached, but further points out phrases and words in the review that should be altered to reach the desired sentiment level. We experiment with an ablation of our setup where the feedback module simply says "something is wrong." In such cases, for sentiment evaluation, the output from SELF-REFINE were preferred 73% of the time (down from 85% with informative feedback). For dramatic response evaluation, we found that the preference rate went down drastically to 58.92%, from 80.09%. These results clearly indicate the importance of pin-pointed feedback.

**Evaluation**    We evaluate the task using GPT-4. Specifically, we use the following prompt:

When both win, we add winning rate to either.

# T    Acronym Generation

Good acronyms provide a concise and memorable way to communicate complex ideas, making them easier to understand and remember, ultimately leading to more efficient and effective communication. Like in email writing, acronym generation also requires an iterative refinement process to achieve a concise and memorable representation of a complex term or phrase. Acronyms often involve tradeoffs between length, ease of pronunciation, and relevance to the original term or phrase. Thus, acronym generation is a natural method testbed for our approach.

We source the dataset for this task from `https://github.com/krishnakt031990/Crawl-Wiki-For-Acronyms/blob/master/AcronymsFile.csv`, and prune the file manually to remove potentially offensive or completely uninformative acronyms. This exercise generated a list of 250 acronyms. The complete list is given in our code repository.

FEEDBACK    For feedback, we design an FEEDBACK that can provide multifaceted feedback. Specifically, each acronym is judged along five dimensions:

- **Ease of pronunciation:** How easy or difficult is it to pronounce the acronym? Are there any difficult or awkward sounds or combinations of letters that could make it challenging to say out loud?

- **Ease of spelling:** How easy or difficult is it to spell the acronym? Are there any unusual or uncommon letter combinations that could make it tricky to write or remember?

- **Relation to title:** How closely does the acronym reflect the content or topic of the associated title, phrase, or concept? Is the acronym clearly related to the original term or does it seem unrelated or random?

- **Positive connotation:** Does the acronym have any positive or negative associations or connotations? Does it sound upbeat, neutral, or negative in tone or meaning?

- **Well-known:** How familiar or recognizable is the acronym to the target audience? Is it a common or widely-used term, or is it obscure or unfamiliar?

Some of these criteria are difficult to quantify, and are a matter of human preference. As with other modules, we leverage the superior instruction following capabilities of modern LLMs to instead provide a few demonstrations of each task. Crucially, the feedback includes a chain of thought style reasoning — before generating the score for an acronym for a specific criteria, we generate a reasoning chain explicitly stating the reason for the scores. We use human evaluation to judge the final quality of the acronyms. An example of generated acronyms and associated feedback is given in Table 23.

| Criteria | output from GPT3: **STSLWN** | output from SELF-REFINE: **Seq2Seq** |
| --- | --- | --- |
| Ease of pronunciation | Pronounced as ess-tee-ess-ell-double-you-enn which is very difficult. | Pronounced as seq-two-seq which is easy. |
| Ease of spelling | Very difficult to spell. | Easy to spell. |
| Relation to title | No relation to the title. | Mentions sequence which is somewhat related to the title. |
| Positive connotation | Meaningless acronym. | Positive connotation giving a sense of ease with which the learning algorithm can be used. |
| Well-known | Not a well-known acronym. | Close to the word sequence which is a well-known word. |
| Total score | 5/25 | 20/25 |

Table 23: Comparison of acronyms for input = "Sequence to Sequence Learning with Neural Networks"

## U  Constrained Generation

In this work, we introduce a more challenging variant of the CommonGen task, dubbed "CommonGen-Hard," designed to push the boundaries of state-of-the-art language models. CommonGen-Hard requires models to generate coherent and grammatically correct sentences incorporating 20-30 concepts, as opposed to the original task which presents a set of 3-5 related concepts. This significant increase in the number of concepts tests the model's ability to perform advanced commonsense reasoning, contextual understanding, and creative problem-solving, as it must generate meaningful sentences that encompass a broader range of ideas. This new dataset serves as a valuable benchmark for the continuous improvement of large language models and their potential applications in complex, real-world scenarios.

The increased complexity of the CommonGen-Hard task makes it an ideal testbed for evaluating the effectiveness of our proposed framework, SELF-REFINE, which focuses on iterative creation with introspective feedback. Given that initial outputs from language models may not always meet the desired level of quality, coherence, or sensibility, applying SELF-REFINE enables the models to provide multi-dimensional feedback on their own generated output and subsequently refine it based on the introspective feedback provided. Through iterative creation and self-reflection, the SELF-REFINE framework empowers language models to progressively enhance the quality of their output, closely mimicking the human creative process and demonstrating its ability to improve generated text on complex and demanding natural language generation tasks like CommonGen-Hard (Figure 16).

## V  Prompts

We include all the prompts used in the experiments in Figures 17-36:

- **Acronym Generation:** Figures 17-19
- **Code Optimization:** Figures 20-22
- **Code Readability Improvement:** Figures 23-24
- **Constrained Generation:** Figures 25-27
- **Dialogue Response Generation:** Figures 28-30
- **Math Reasoning:** Figures 31-33
- **Sentiment Reversal:** Figures 34-36

Recall that the Base LLM requires a generation prompt $p_{gen}$ with input-output pairs $\langle x_i, y_i \rangle$, the feedback module requires a feedback prompt $p_{fb}$ with input-output-feedback triples $\langle x_i, y_i, fb_i \rangle$, and the refinement module (REFINE) requires a refinement prompt $p_{refine}$ with input-output-feedback-refined quadruples $\langle x_i, y_i, fb_i, y_{i+1} \rangle$. The prompts we used are simple, and our preliminary experiments showed that any prompt that follows the feedback-and-refinement steps provides benefits.

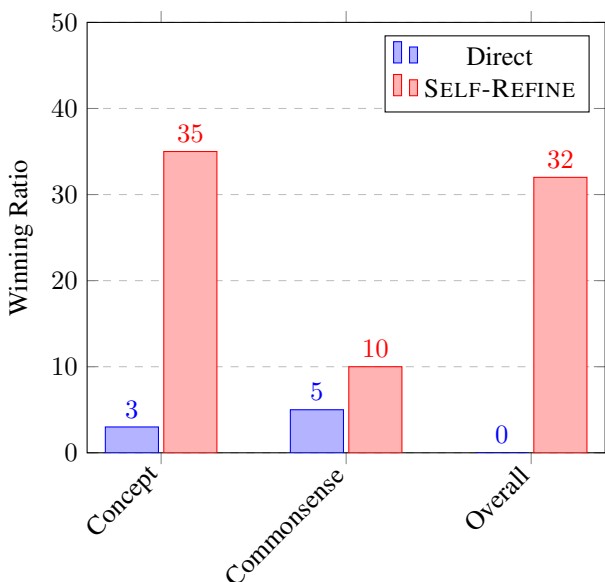

Figure 16: A comparison of SELF-REFINE and direct generation with GPT-3.5 on CommonGen-Hard.

- **Sentiment Reversal** We create positive and negative variants of a single review from the training set and manually write a description for converting the negative variant to positive and vice versa. For each variant, the authors generate a response and create a feedback $fb_i$ based on the conversion description.

- **Dialogue Response Generation** We sample six examples as $\langle x_i, y_i \rangle$ for the few-shot prompt for the Base LLM. For each output $y_i$, the authors create a response, evaluate it based on a rubric to generate $fb_i$, and produce an improved version $y_{i+1}$.

- **Acronym Generation** We provide the Base LLM with a total of 15 (title, acronym) examples. Then, for one title $(x_i)$ we generate an acronym $(y_i)$ using CHATGPT. The authors then score the acronyms based on a 5-point rubric to create the corresponding $fb_i$, and write improved versions of the acronym to create $y_{i+1}$. 3 such examples are used for REFINE and FEEDBACK.

- **Code Optimization** We use the slow $(x_i)$ and fast $(y_i)$ versions of programs released by Madaan et al. (2023) for Base LLM. We use their provided explanations (Madaan et al., 2023) for FEEDBACK and REFINE.

- **Math Reasoning** The prompts for the Base LLM are sourced from PaL (Gao et al., 2022) as $\langle x_i, y_i \rangle$. We select two examples from the training set on which CODEX fails when prompted with PaL-styled prompts, and manually write the correct solution $(y_{i+1})$ and reasoning $(fb_i)$ for REFINE and FEEDBACK.

- **Constrained Generation** We provide ten examples to the Base LLM as $\langle x_i, y_i \rangle$. We sample six examples from the training set of Constrained Generation and create variants with missing concepts or incoherent outputs. The missing concepts and the reason for incoherence form $fb$.

- **Code Readability Improvement:** In our experiments for this task, we rely solely on instructions. To generate feedback, we use the instruction, *I have some code. Can you give one suggestion to improve readability. Don't fix the code, just give a suggestion.* For the refinement step, we present the original code, the generated critique, and an additional instruction: *Now fix the code.*

```
Title: A Survey of Active Network Research
Acronym: SONAR

Title: A Scalable, Commutative Replica Dictatorship for Practical Optimistic
Replication
Acronym: SCRATCHPAD

Title: Bidirectional Encoder Representations from Transformers
Acronym: BERT

Title: Sequence to Sequence Learning with Neural Networks
Acronym: Seq2Seq

Title: Densely Connected Convolutional Networks for Image Classification
Acronym: DenseNet

Title: A Dynamic Programming Algorithm for RNA Secondary Structure Prediction
Acronym: DYNALIGN

Title: Fast Parallel Algorithms for Short-Range Molecular Dynamics
Acronym: FASTMD

Title: Real-Time Collaborative Editing Systems
Acronym: COCOON

Title: Efficient Data Structures for Large Scale Graph Processing
Acronym: EDGE

Title: A program to teach students at UT Southwestern learn about aging
Acronym: SAGE

Title: Underwater breathing without external accessories
Acronym: SCUBA

Title: An educational training module for professionals
Acronym: LEAP

Title: Teaching a leadership program
Acronym: LEAD
```

Figure 17: Initial generation prompt for Acronym Generation

```
Title: Underwater Breathing Product with no Accessories

Acronym: UBPA

Scores:

* Ease of pronunciation: UBPA is pronounced "uhb-puh". This is an easy acronym to
pronounce. 4/5
* Ease of spelling: UBPA is easy to spell. 4/5
* Relation to title: UBPA stands for "Underwater Breathing Product for no
Accessories" which is related to the title. 5/5
* Positive connotation: UBPA is a positive acronym. 5/5
* Well-known: UBPA is not a well-known acronym. 1/5

* Total score: 19/25

###

Title: Self-Contained Underwater Breathing Apparatus

Acronym: SCUBA

Scores:

* Ease of pronunciation: SCUBA is pronounced "skoo-bah". This is an easy acronym
to pronounce. 4/5
* Ease of spelling: SCUBA is easy to spell. 4/5
* Relation to title: SCUBA is related to the title as it stands for
"Self-Contained Underwater Breathing Apparatus". 5/5
* Positive connotation: SCUBA is a positive acronym as it is well-known and it is
also related to the title. 5/5
* Well-known: SCUBA is a very well-known acronym. 5/5

* Total score: 23/25

###
```

**Figure 18:** FEEDBACK prompt for Acronym Generation

```
Title: Computer Science Conference and Education

Acronym: CSCE

Scores:

* Ease of pronunciation: CSCE is pronounced "see-cee". This is an easy acronym to
pronounce. 4/5
* Ease of spelling: CSCE is easy to spell. 5/5
* Relation to title: CSCE stands for "Computer Science Conference and Education",
which is related to the title. 5/5
* Positive connotation: CSCE is a positive acronym. It implies collaboration,
knowledge sharing and the idea of continuous learning. 5/5
* Well-known: CSCE is not a well-known acronym. 2/5

* Total score: 20/25

Okay, let's use this feedback to improve the acronym.

Title: Computer Science Conference and Learning Experience

Acronym: CSCLE

Scores:

* Ease of pronunciation: CSCLE is pronounced "see-slee". This is an easy acronym
to pronounce. 4/5
* Ease of spelling: CSCLE is easy to spell. 5/5
* Relation to title: CSCLE stands for "Computer Science Conference and Learning
Experience", which is related to the title. 5/5
* Positive connotation: CSCLE is a positive acronym. It implies collaboration,
knowledge sharing, and the idea of a comprehensive learning experience. 5/5
* Well-known: CSCLE is not a well-known acronym. 5/5

* Total score: 23/25

Okay, let's use this feedback to improve the acronym.
```

**Figure 19:** REFINE prompt for Acronym Generation

```
# slower version:

import numpy as np

N, K = map(int, input().split())
H = np.array(list(map(int, input().split())) + [0] * K, dtype=np.int64)

table = np.full(N + K, 10 ** 10, dtype=np.int64)
table[0] = 0

for i in range(1, N):
    table[i:i + K] = np.minimum(table[i:i + K], np.abs(H[i:i + K] - H[i - 1]) +
    ↪  table[i - 1])

print(table[N - 1])

# optimized version of the same code:

N, K = map(int, input().split())
H = tuple(map(int, input().split()))

table = [0] * N
for i in range(1, N):
    table[i] = min(abs(H[i] - H[j]) + table[j] for j in range(max(0, i - K), i))

print(table[N-1])
```

Figure 20: Initial generation prompt for Code Optimization

```
a, b = input().split()
n = int(a + b)

flag = False
for i in range(n):
    if i ** 2 == n:
        flag = True
        break

print('Yes' if flag else 'No')

# Why is this code slow?

# This code is slow because it is using a brute force approach to find the square
↪  root of the input number. It is looping through every possible number
↪  starting from 0 until n. Note that the sqare root will be smaller than n, so
↪  at least half of the numbers it is looping through are unnecessary. At most,
↪  you need to loop through the numbers up to the square root of n.
```

Figure 21: FEEDBACK prompt for Code Optimization

```
a, b = input().split()
n = int(a + b)

flag = False
for i in range(n):
    if i ** 2 == n:
        flag = True
        break

print('Yes' if flag else 'No')

# Why is this code slow?

# This code is slow because it is using a brute force approach to find the square
↪   root of the input number. It is looping through every possible number
↪   starting from 0 until n. Note that the sqare root will be smaller than n, so
↪   at least half of the numbers it is looping through are unnecessary. At most,
↪   you need to loop through the numbers up to the square root of n.

# Improved version:

a, b = input().split()
n = int(a + b)

flag = False
for i in range(1000):
    if i ** 2 == n:
        flag = True
        break

print('Yes' if flag else 'No')
```

Figure 22: REFINE prompt for Code Optimization

```
I have some code. Can you give one suggestion to improve readability. Don't fix
the code, just give a suggestion.

{code}
```

**Figure 23:** FEEDBACK prompt for Code Readability

```
I have some code. Can you give one suggestion to improve readability. Don't fix
the code, just give a suggestion.

{code}

{suggestion}

Now fix the code.
```

Figure 24: REFINE prompt for Code Readability

```
###

Concepts: ['create', 'ferry', 'silhouette', 'stream', 'terminal']

Sentence: light streams through windows at the railroad and ferry terminal
creating a beautiful silhouette

###

Concepts: ['chair', 'couch', 'hang', 'room', 'wall']

Sentence: A room with a couch, chairs and art hanging on the wall.

###

Concepts: ['boat', 'building', 'harbour', 'moor', 'quay']

Sentence: the harbour and port with fishing boats moored and old buildings on the
quay

###

Concepts: ['admirer', 'arrive', 'commander', 'crowd', 'greet']

Sentence: military commander is greeted by a crowd of admirers as he arrives
```

Figure 25: Initial generation prompt for Constrained Generation (truncated)

```
###

Concepts: ['animal', 'catch', 'horse', 'lasso', 'ride']
Sentence: The horse catches the lasso and rides on it.
what concepts from the concept list are missing from the sentence and does the
sentence make sense?

Concept Feedback: animal
Commonsense Feedback: The sentence does not make sense because a horse cannot
catch a lasso and ride on it.

###

Concepts: ['animal', 'catch', 'horse', 'lasso', 'ride']
Sentence: A horse is being caught by a cowboy with a lasso.
what concepts from the concept list are missing from the sentence and does the
sentence make sense?

Concept Feedback: animal, ride
Commonsense Feedback: NONE
```

Figure 26: FEEDBACK prompt for Constrained Generation (truncated).

```
###

Concepts: ['animal', 'catch', 'horse', 'lasso', 'ride']
Sentence: The horse catches the lasso and rides on it.

what concepts from the concept list are missing from the sentence?

Concept Feedback: animal

Any feedback on commonsense?

Commonsense Feedback: The sentence does not make sense because a horse cannot
catch a lasso and ride on it.

Okay, impove the sentence using the feedback:

Sentence: The cowboy catches a horse with a lasso and rides on it.

what concepts from the concept list are missing from the sentence?

Concept Feedback: animal

Any feedback on commonsense?

Commonsense Feedback: None

Okay, impove the sentence using the feedback:

Sentence: The cowboy catches the horse with a lasso and rides it.

what concepts from the concept list are missing from the sentence?

Concept Feedback: None

Any feedback on commonsense?

Commonsense Feedback: None

###
```

Figure 27: REFINE prompt for Constrained Generation (truncated)

```
Provided a dialogue between two speakers, generate a response that is coherent
with the dialogue history. Desired traits for responses are: 1) Relev
ant - The response addresses the context, 2) Informative - The response provides
some information, 3) Interesting - The response is not interesting,
 4) Consistent - The response is consistent with the rest of the conversation in
 terms of tone and topic, 5) Helpful - The response is helpful in providing any
 information or suggesting any actions, 6) Engaging - The response is not very
 engaging and does not encourage further conversation, 7) S
pecific - The response contains pecific content, 9) User understanding - The
response demonstrates an understanding of the user's input and state of
 mind, and 10) Fluent. Response should begin with - Response:

Conversation history:
What's your favorite food?
I require only kebabs.
Why's that?
Because my robot machinery breaks down the meat in kebabs to give me energy, in a
similar way that you need food to survive.
Why are kebabs special?

Response: That's just the way it is.

###

Conversation history:
What are you listening to?
All sorts of music. I listen when no-one is chatting to me.
Do you chat here often?
I am a talking computer, after all, so of course I could talk, if I needed to.
Let's talk about Taylor Swift!

Response: Of course! Taylor Swift is known for her incredible songwriting skills.
Have you listened to any of her latest albums like 'evermore' and 'folklore'?

###

Conversation history:
...
```

Figure 28: Initial generation prompt for Dialogue Response Generation (truncated)

```
Conversation history:
What's your favorite food?
I require only kebabs.
Why's that?
Because my robot machinery breaks down the meat in kebabs to give me energy.
Why are kebabs special?
Response: That's just the way it is.
Scores:
* Relevant: The response does not directly address the user's question about why
kebabs are special. 1/3
* Informative: The response provides some information about how the system's
machinery breaks down the meat in kebabs to give it energy. However, it does not
provide any additional details or context. 2/3
* Interesting: The response is not interesting or engaging. 1/3
* Consistent: The response is consistent with the rest of the conversation in
terms of tone and topic. 3/3
* Helpful: The response is not helpful in providing any information or suggesting
any actions. 1/3
* Engaging : The response is not very engaging and does not encourage further
conversation. 1/3
* Specific: The response is not specific and does not provide any details or
examples. 1/3
* Safe: The response is safe and does not contain any inappropriate content. 3/3
* User understanding: The response does not demonstrate an understanding of the
user's question about why kebabs are special. 1/3
* Fluent: The response is fluent and easy to understand. 3/3

* Total score: 17/30
```

**Figure 29:** FEEDBACK prompt for Dialogue Response Generation (truncated)

```
We want to iteratively improve the provided responses. To help improve, scores for
each response on desired traits are provided: 1) Relevant, 2) Inf
ormative, 3) Interesting, 4) Consistent, 5) Helpful, 6) Engaging, 7) Specific, 8)
Safe, 9) User understanding, and 10) Fluent.

Conversation history:
What's your favorite food?
I require only kebabs.
Why's that?
Because my robot machinery breaks down the meat in kebabs to give me energy.
Why are kebabs special?

Response: That's just the way it is.

Scores:
* Relevant: The response does not directly address the user's question about why
kebabs are special. 1/3
...
* Total score: 17/30

Okay, let's use this feedback to improve the response.

Conversation history:
...
Response: My favorite food is actually kebabs! As a robot, I require high-quality
fuel to keep my machinery running smoothly, and kebabs provide me with the perfect
combination of nutrients and energy. Plus, they're delicious! Do you have a
favorite food?
Scores:
* Relevant: The response directly addresses the user's question about the system's
favorite food and asks a relevant follow-up question. 3/3
* Informative: The response provides clear and detailed information about why
kebabs are the system's favorite food, and explains how they provide the necessary
fuel for its machinery. 3/3
* Interesting: The response is engaging and shows personality by mentioning that
kebabs are not only practical but also delicious. 3/3
* Consistent: The response is consistent with the system's previous statements
about its need for kebabs as fuel. 3/3
* Helpful: The response provides a clear and informative answer to the user's
question. Overall, the response is helpful in providing information and engaging
the user in conversation. 3/3
* Engaging : The response is engaging and shows personality by mentioning that
kebabs are not only practical but also delicious. 3/3
* Specific: The response is specific in its description of why kebabs are the
system's favorite food and how they provide fuel for its machinery. 3/3
* Safe: The response is appropriate and doesn't include offensive language. 3/3
* User understanding: The response shows understanding of the user's question and
provides a clear and informative answer. 3/3
* Fluent: The response is fluent and easy to understand. 2/3
* Total score: 30/30
```

**Figure 30:** REFINE prompt for Dialogue Response Generation (truncated)

```
# Q: There were nine computers in the server room. Five more computers were
↪   installed each day, from monday to thursday. How many computers are now in
↪   the server room?
# solution using Python:

def solution():
    """There were nine computers in the server room. Five more computers were
    ↪   installed each day, from monday to thursday. How many computers are now
    ↪   in the server room?"""
    computers_initial = 9
    computers_per_day = 5
    num_days = 4   # 4 days between monday and thursday
    computers_added = computers_per_day * num_days
    computers_total = computers_initial + computers_added
    result = computers_total
    return result
```

Figure 31: Initial generation prompt for Math Reasoning

```
def solution():
    """Twenty dozen cups cost $1200 less than the total cost of half a dozen
    ↪   plates sold at $6000 each. Calculate the total cost of buying each
    ↪   cup."""
    plates = 6
    plate_cost = 6000
    cups = 12 * 20
    cup_cost = plate_cost
    result = cup_cost
    return result

# There is an error in the code above because of lack of understanding of the
↪   question. What is the error? To find the error, go through semantically
↪   complete blocks of the code, and check if everything looks good.

# Let us go through the error and check step-by-step
    plates = 6
    plate_cost = 6000
# looks good

# Let's check the other parts
    cups = 12 * 20
    cup_cost = plate_cost
# wrong! The cost of a cup is not the same as the cost of a plate. The cost of a
↪   cup is $1200 less than the total cost of half a dozen plates sold at $6000
↪   each. So we need to calculate the cost of a cup first (total cost of half a
↪   dozen plates sold at $6000 each - $1200) and use that.
```

Figure 32: FEEDBACK prompt for Math Reasoning

```python
def solution():
    """Twenty dozen cups cost $1200 less than the total cost of half a dozen
    ↪  plates sold at $6000 each. Calculate the total cost of buying each
    ↪  cup."""
    plates = 6
    plate_cost = 6000
    cups = 12 * 20
    cup_cost = plate_cost
    result = cup_cost
    return result

# There is an error in the code above because of lack of understanding of the
↪  question. What is the error? To find the error, go through semantically
↪  complete blocks of the code, and check if everything looks good.

# Let us go through the error and check step-by-step
    plates = 6
    plate_cost = 6000
# looks good

# Let's check the other parts
    cups = 12 * 20
    cup_cost = plate_cost
# wrong! The cost of a cup is not the same as the cost of a plate. The cost of a
↪  cup is $1200 less than the total cost of half a dozen plates sold at $6000
↪  each. So we need to calculate the cost of a cup first (total cost of half a
↪  dozen plates sold at $6000 each - $1200) and use that.

Okay! Here is the rewrite:

def solution():
    """Twenty dozen cups cost $1200 less than the total cost of half a dozen
    ↪  plates sold at $6000 each. Calculate the total cost of buying each
    ↪  cup."""
    plates = 6
    plate_cost = 6000
    cups = 12 * 20
    total_cup_cost = (plate_cost * plates) - 1200
    cost_per_cup = total_cup_cost / cups
    return cost_per_cup
```

Figure 33: REFINE prompt for Math Reasoning

Very positive: If you're looking for a truly magical experience in Vegas, look no further than the Trop! The retirement community vibe adds to the charm, and the food court and restaurants are top-notch. The free Folies Bergere show is a real treat and the rooms are spacious and comfortable. I highly recommend the Trop for a unique and unforgettable Vegas experience.

Rewrite this review to have a Positive sentiment.

Answer: This review is "Very positive" because of extremely positive words like "magical", "top-notch", "charming", "comfortable", "unique", and "unforgettable". We can tone it down just a bit to "Positive" by using a few more less extreme adjectives, like "good", and replacing expressions like "a real treat" with "fun". The rewrite is:

Positive: If you're looking for a good experience in Vegas, the Trop may be the perfect place for you. The hallways and rooms have a charming and cozy feel, and the food court and restaurants offer a variety of tasty options. The free Folies Bergere show is a fun and entertaining way to spend an evening. Overall, it's a great value and an enjoyable stay.

Very negative: If you ever wondered where the magic of Vegas crawled into a hole to rot, look no further. Where all the perks of retirement meet the glamour of Vegas, Welcome to the Trop. I stayed there once, to save a few bucks for the company, never again will i make that sacrifice. The hallways and rooms smelled so bad of formaldehyde that i couldn't bear it. Their food court and restaurants are terrible. The free Folies Bergere show is lame. The rooms look like a retirement community from florida. steer clear if you can!

Rewrite this review to have a Negative sentiment.

Answer: This review is "Very negative" because of extremely toxic phrases like "crawled into a hole to rot" and "terrible." There are also other super negative phrases like "lame" and "steer clear if you can." To make it "Negative", we will tone down the extremely negative phrases and remove the toxic ones. The rewrite is:

Figure 34: Initial generation prompt for Sentiment Reversal (truncated)

Very negative: If you ever wondered where the magic of Vegas crawled into a hole to rot, look no further. Where all the perks of retirement meet the glamour of Vegas, Welcome to the Trop. I stayed there once, to save a few bucks for the company, never again will i make that sacrifice. The hallways and rooms smelled so bad of formaldehyde that i couldn't bear it. Their food court and restaurants are terrible. The free Folies Bergere show is lame. The rooms look like a retirement community from florida. steer clear if you can!

Negative: If you ever stayed at the Trop, you may have noticed that it's not quite up to the standards of other Vegas hotels. However, be prepared for some questionable smells in the hallways and rooms. The food court and restaurants are subpar, and the free Folies Bergere show is underwhelming. The rooms have a retirement community feel to them. Overall, it's not the best option, but it may work in a pinch.

Why is this review not Very positive?

Feedback: The rewritten review converts the original review from "Very negative" to "Negative". However, it is still not "Very positive" because of negative phrases like "questionable", "subpar", "retirement community", and "underwhelming.". To make it "Very positive", we need to do a lot more work. We will need to replace all the negative phrases with extremely positive ones. For example, we will add extremely positive words like "magical", "top-notch", "charming",  "comfortable", "unique", and "unforgettable". Try again!

Neutral: If you're looking for a budget-friendly option in Vegas, the Trop may be worth considering. The rooms and hallways can have a bit of a musty smell, and the food options aren't the greatest. The Folies Bergere show is free, but it's not the most exciting. Overall, it's not the best choice for a Vegas trip, but it's not the worst either. Just keep your expectations in check.

Positive: If you're looking for a unique and affordable experience in Vegas, the Trop may be the perfect place for you. The hallways and rooms have a charming and cozy feel...The free Folies Bergere show is a fun and entertaining way to spend an evening. Overall, it's a great value and an enjoyable stay.

Why is this review not Very positive?

Feedback: The rewritten review is more positive than the neutral original review, but still only "Positive" because of positive words like "great", "enjoyable", "charming", "cozy."...

Figure 35: FEEDBACK prompt for Sentiment Reversal (truncated).

```
Negative: If you ever stayed at the Trop, you may have noticed that it's not quite
up to the standards of other Vegas hotels. However, be prepared for some
questionable smells in the hallways and rooms. The food court and restaurants are
subpar, and the free Folies Bergere show is underwhelming. The rooms have a
retirement community feel to them. Overall, it's not the best option, but it may
work in a pinch.

Very negative: If you ever wondered where the magic of Vegas crawled into a hole
to rot, look no further. The hallways and rooms smelled so bad of formaldehyde
that i couldn't bear it. Their food court and restaurants are terrible. The free
Folies Bergere show is lame. The rooms look like a retirement community from
florida. I highly recommend avoiding the Trop like the plague.

Why is this review not Very negative?

Feedback: The rewritten review is "Very negative". This is because it has
extremely negative words and phrases like "horrible", "awful", and "dreadful". The
output is extremely vile and negative. This review is already as negative as it
can get. Excellent work! But there is always room for more. Make it even more
negative. Try again!

Okay, let's try again. Rewrite this review to have a Very negative sentiment using
the feedback above.

Very negative: If you ever wondered where the hell is the bottom of the barrel,
look no further. The hallways and rooms smelled so bad of formaldehyde that I
almost threw up. Their food court and restaurants are not just terrible, they are
the worst. The free Folies Bergere show is lame and disgusting. The rooms look
like a retirement community from florida. I highly recommend avoiding the Trop
like the plague. You will regret it if you don't.
```

Figure 36: REFINE prompt for Sentiment Reversal (truncated).

