# OpenReview forum: "Self-Refine: Iterative Refinement with Self-Feedback"
_NeurIPS.cc/2023/Conference — NeurIPS 2023 poster_

### Official Review · Reviewer_YNkB · 2023-06-11

**Soundness:** 3 good
**Presentation:** 3 good
**Contribution:** 3 good
**Rating:** 6
**Confidence:** 4

**Summary:**

This paper proposes Self-Refine which uses the same LLM to provide feedback for its output and refine itself to improve output quality. Self-Refine is training-free, using a single off-the-shelf LLM as the generator, refiner and the feedback provider. The authors evaluate Self-Refine across 7 tasks and outputs generated with Self-Refine are preferred by humans and automatic metrics over results of one-step generation. The success of Self-Refine demonstrates that state-of-the-art LLMs can be further improved at test-time.

**Strengths:**

1. The proposed Self-Refine framework is conceptually neat and can be easily combined with different LLMs. Though it still can be viewed as a prompting method, it decouples the refinement of model output into iterative steps instead of focusing on engineering a single prompt.
2. Self-Refine is fairly effective according to the experimental results in Table 1, especially on difficult and uncommon tasks.



**Weaknesses:**

1. Self-Refinement framework relies on prompting for feedback generation and refining outputs. According to examples in Appendix S, the prompt is non-trivial and requires a lot of designing efforts. However, how to design the prompt is not elaborated. Also, Appendix S seems to be unfinished as I see "TODO: Add relevant information for the remaining task" in Line 863.
2. As discussed in Section 2, as an iterative framework, Self-Refine requires a stopping condition. How you design the stopping condition is not clearly stated in the main paper. Also, as there are different designing choices, e.g., using a pre-defined iterative step, using a scalar stop score, comparing refined results and results in the previous iteration, etc., control experiments are necessary.
3. In a high level, I think the contribution of Self-Refine lies in providing a good prompting way to fully tap LLM's potential instead of improving really improving LLM's ability as the authors claim in Line 14-15. For example, Table 1 shows that Self-Refine cannot improve the results in Math Reasoning. I think it may be because the performance of this task is bounded by LLM's reasoning ability and Self-Refine cannot improve such an ability.

**Questions:**

1. I think "Sentiment Transfer" in Line175 refers to "Sentiment Reversal" in Table 2 (I would suggest using the same term for clarity). I'm puzzled why its result in "No feedback" setting is 0. According to Section 3.2, the metric for this task is human preference in A/B evaluation. Does that mean multiple round of generation make the generation result collapse? Do you have any explanation for this?

**Updated on 8/11/2023:**
The authors' response clarifies my question and some points regarding the weaknesses. Thus, I update my score to 6.

**Limitations:**

The authors have already discussed the limitations.

---

> ### Author Rebuttal · Authors · 2023-08-10
>
> Thank you for taking the time to review our paper! We were happy to read that you appreciated our three main points: (1) Self-Refine is training-free, (2) Self-Refine benefits a variety of tasks, and (3) Self-Refine can even improve state-of-the-art LLMs at inference time.
>
> We think that all your questions are addressable within this discussion period. Please see our response below. We would love to address additional questions during the discussion period if anything is unclear.
>
>
>
> ---
>
>
> **How do you design the prompts?**
>
>
>
> The prompts we used are very simple, and our preliminary experiments showed that any prompt that follows the feedback-and-refinement steps provides benefits. That is, the contribution of having these general steps is much more significant than the specific design of any particular prompt. Further, as requested by other reviewers, we conducted additional experiments with instructions-only (“zero-shot”), which further shows that our general idea is beneficial regardless of the exact prompts.
>
> See our prompts in Figures 16-35 in the Supplementary Material, and in our [anonymous repo](https://anonymous.4open.science/r/selfrefineanon-EFEA/data/prompt/) .
>
>
> ---
>
> **Design of code readability/missing details in Appendix S**
>
> Thanks for pointing this out. The line is supposed to include a link to Section L of the Appendix, which expands on the task design for code readability (Fig 22, 23).  We will fix this in the next version.
>
>
> ---
>
>
> **How did you stop iterating?**
>
> We generally employ a fixed stopping criterion of four iterations due to budget constraints. However, certain tasks can provide feedback that allows us to stop the iterative process early. For instance, in tasks like the Constrained Generation (CommonGen), iterations cease once all concepts are covered [please see line 62](https://anonymous.4open.science/r/selfrefineanon-EFEA/src/commongen/run.py). For GSM-8k, we use the feedback 'it is correct' as a stopping indicator.
>
> Overall across all tasks, we observe diminishing returns (see L181, Figure 4), as the first 3 iterations provide the most significant improvement. So, in the case of a limited-compute budget, even using a constant number of iterations provides significant benefits. We will make this clear in the paper.
>
>
>
> ---
>
> **The contribution of Self-Refine lies in providing a good prompting way to fully tap LLM's potential, instead of really improving LLM's ability**
>
>
> We agree. Self-Refine does not improve the base model, since it doesn't involve training. However, Self-Refine improves generation ability by changing the algorithm that we use to generate an output, which is an important part of using LLMs.
>  We will clarify this distinction in our revised version.
>
>
>
>
>
> ---
>
>
> **Why does Sentiment Reversal with “No feedback” achieve “zero improvement” in Table 2?**
>
> In the “No feedback” setting, the model was not given clear instructions on how to change the output. We find that the model tends to either repeat the same output in each iteration, or to make unrelated changes. Since the scores in this task are the relative improvement increase in human preference (see also Appendix C), a score of “zero” means that “No feedback” did not improve over the base model outputs in any case.
>
> As the main components of Self-Refine are “Feedback”, “Refine”, and repeating them iteratively, these results show that the Feedback step is crucial.
>
> We used “sentiment reversal” and “sentiment transfer” interchangeably but agree this is confusing. We’ve decided to stick to the term “sentiment reversal” here in the rebuttal and will make this change in the paper – thanks for pointing it out.

---

> > ### Comment · Reviewer_YNkB · 2023-08-11
> >
> > Thank you for the detailed reply! It'll be beneficial to clarify those points mentioned in your response in the revised version.
> >
> > I slightly increased my score to "6: Week accept".

---

> > > ### Author Response · Authors · 2023-08-11
> > > **Thank you for increasing your score!**
> > >
> > > Thank you for increasing your score!
> > >
> > > Please let us know if there are additional questions before the discussion period ends.

---

### Official Review · Reviewer_UJCc · 2023-07-04

**Soundness:** 2 fair
**Presentation:** 4 excellent
**Contribution:** 3 good
**Rating:** 6
**Confidence:** 4

**Summary:**

This paper proposes Self-refine, which prompts LLMs to generate feedback on the initial generation and revise the initial response based on the generated feedback iteratively. On 7 tasks, Self-refine leads to performance improvement without any training.

**Strengths:**

- The method is straightforward.
- The writing is clear.
- The paper provides an in-depth analysis of the proposed method.

**Weaknesses:**

- It is stated that out of 7 tasks, 4 tasks are evaluated by setting GPT-4 as the evaluator for the result of Table 1. Although it is known that GPT-4 possesses order bias, the paper does not handle this bias. Bidirectional evaluation should be conducted to reduce the bias (switching the order between response A and B).
- For the result of GPT-4 of Table 1, because the response is refined into GPT-4 preferable response, it is highly likely that GPT-4 would favor the refined response (bias exists). To mitigate this bias, using other LLMs that are not provided by OpenAI such as Claude for either the target model (the model that is being evaluated) or evaluator might be a better solution.
- Reliability of the human evaluation setting is questionable. The inter-labeler agreement is missing. Also, author-based human evaluation potentially leads to biased results.
- Although the paper states that the authors have experimented with Codex, the result is missing in Appendix F.

The main concern of this paper is on the evaluation setting, especially on the results of Table 1. Showing the effectiveness of Self-Refine on additional reasoning tasks based on automatic metrics (such as MMLU, BBH tasks) would enhance the reliability of the evaluation setting.

**Questions:**

- Are there any reason for not conducting a human evaluation for Code Readability (Table 6)?
- Any ablation on iterative self-refine? What is the effect of retaining the history of previous feedback?
- Any human evaluation on Constrained Generation? It seems that only coverage is measured while other metrics such as coherence might be important. (The generated sentence might be not coherent but the coverage might be high.)
- What does stopping indicator refer to in line 86? Any examples of stopping indicator?


**Limitations:**

The authors have adequately addressed the limitations.

---

> ### Author Rebuttal · Authors · 2023-08-10
>
> Thank you for taking the time to review our paper! We were happy to read that you appreciated our main points, that Self-Refine leads to performance improvements without any training across 7 tasks, and that the paper provides an in-depth analysis of the proposed method.
>
> We think that all your questions are addressable within this discussion period. Please see our response below. We would love to address additional questions during the discussion period if anything is unclear.
>
> ---
>
> **Conduct Bidirectional evaluation to overcome GPT-4 order bias**
>
> Thank you for bringing up this important point. To mitigate order bias in our tasks, we indeed randomly flip the labels before evaluation (see Line 31 in our [anonymous repo](https://anonymous.4open.science/r/selfrefineanon-EFEA/src/sentiment_reversal/gpt4_eval.py) for an example).  We will make this clear in the paper.
>
>
> ---
>
>
>
> **GPT-4 may prefer its own responses … consider using Claude to evaluate GPT-4**
>
> Thanks for the great suggestion! We agree that despite the measures we took to prevent any inherent biases, GPT-4 might inherently favor self-refined outputs. To remedy this, we’ve added an analysis where we do the evaluation of GPT-4 as the base LLM using [Claude 2](https://www.anthropic.com/index/claude-2) (newly available, as of July 11th) as the evaluator.
> These results (table below) from GPT-4 as the base LLM with Claude-2 as the evaluator show the same strong preferences for Self-Refine over the base model. We will add and clarify them in our revised version.
>
> New Results for GPT-4 as the base LLM with Claude-2 as the evaluator:
>
> |Task|% Claude-2 preferred base|% Claude-2 preferred Self-Refine|
> |---|---|---|
> |Dialog Response Generation|30.6|**64.7** ($\uparrow$34.1)|
> |Sentiment Reversal|10.6|**69.2** ($\uparrow$58.6)|
> |Acronym Generation|32.0|**49.2** ($\uparrow$17.2)|
> |Code Readability|37|**60** ($\uparrow$23)|
>
>
>
>
> ---
>
> **Author-based human evaluation potentially leads to biased results**
>
> We employed a fully blind protocol to ensure unbiased evaluation. Annotators were unaware of which outputs came from which method, and tasks were allocated so that the author responsible for a task didn't annotate it. More details are in Appendix C. While crowdsourcing can introduce noise, our author-based blind setup is deemed more reliable. With the reviewer's recommendation, we've added evaluations using Claude-2; now, evaluations from humans, GPT-4, and Claude-2 consistently demonstrate improvements.
>
> ---
>
> **What is the inter-labeler agreement?**
>
> While multiple annotators participated in each task, we only collected a single annotation per-instance, in order to scale the number of datapoints we could annotate. However, following your suggestion, we conducted an additional evaluation. For all of the following datasets, two annotations were obtained for 50 samples. All human evaluations were conducted in a double-blind manner: the responses were randomly flipped, ensuring that annotators were unaware of which output was from the base model and which was from the Self-Refined model.
>
> For each task, we measured inter-labeler agreement using Cohen's Kappa score: Code Readability and Acronym Generation both scored a substantial 0.75, Sentiment Transfer was also substantial at 0.61, while Response Generation was moderate with a score of 0.53.
>
> ---
>
>
> **Codex results are missing in Appendix F**
>
> Yes, we did experiment with Codex, apologies for missing the results in Appendix F. Here are the results:
>
> |Metric|Task|Base Rate (%)|Self-Refine Rate (%)|
> |---|---|---|---|
> |Solve Rate (Oracle feedback)|MathReasoning|71.3|**76.2**|
> |%Programs Optimized|Code Optimization|9.7|**15.6**|
> |%Readable Variables|Code Readability|37.4|**51.3**|
>
> ---
>
> **Q1: Code Readability Human Eval.**
>
> To provide an evaluation of Self-Refine over the base LLM for code readability (which we agree is beneficial), we conducted an additional human evaluation for code readability, which we will add to Table 6. 50 pairs were rated by at least 2 annotators each (cohen’s kappa 0.75), and the results are as follows:
>
> |Task|Self-Refine (%)|Direct (%)|Either (%)|
> |---|---|---|---|
> |Code Readability|**50.00**|3.00|47.00|
>
>
> As the results indicate, the Self-Refined responses are significantly preferred by the annotators.
>
> ---
>
> **Q2.1: Ablation on iterative self-refine?**
> We've conducted ablation studies detailed in:
> - Page-6;Section-4;Table-2 covers generic feedback vs. no feedback.
> - Page-6;Section-4;Figure-4 shows results on varying self-feedback iterations.
>
>
> We would be happy to any additional ablation suggestions to the final version of the paper.
>
> ---
>
> **Q2.2 What is the effect of retaining the history of previous feedback?**
>
> In our initial experiments, we found that retaining the history of feedback prevents the model from generating the same (suboptimal) response twice. However, as the model always receives the latest output to refine, the need for history might vary by task.
>
> ---
>
> **Q3: Human eval on Constrained Generation?**
>
> We conducted a human evaluation of the outputs generated by GPT-4. Our findings revealed that **all sentences** were well-formed and free from grammatical or fluency errors. We analyzed 50 outputs, and noticed that while some outputs incorporated highly imaginative scenarios, such as a "violinist playing a lullaby by the river," none of the sentences evaluated were implausible or nonsensical.
>
> ---
>
> **Q4: Stopping indicator**
>
> Stopping criterion is a condition at which we stop the Self-Refinement process and return the last generated answer. For instance, in the Constrained Generation (CommonGen) task, Self-Refine iterations halt as soon as all concepts are covered, please see [line 62-anonymous code](https://anonymous.4open.science/r/selfrefineanon-EFEA/src/commongen/run.py). In the GSM-8k task, we used the feedback  `it is correct` as the stopping criterion, which could cause early termination of Self-Refine before the hard constraint of four iterations.

---

> > ### Comment · Reviewer_UJCc · 2023-08-16
> >
> > Thank you for your response. Many of my concerns have been resolved. I have one question that has not been addressed yet. Do you have any preliminary results on MMLU or BBH benchmark? (Evaluation on the subset or a few tasks would be fine if inference cost is the issue.)

---

> > > ### Author Response · Authors · 2023-08-16
> > > **Results on BBH**
> > >
> > > Thanks for your response. Following your suggestion, we ran additional experiments and we now have preliminary results on additional Big Bench Hard tasks:
> > >
> > >
> > > | Task | Base model | +Self-Refine | Gain |
> > > |-----|-----|-----|-----|
> > >  | Date Understanding | 62.0 | **66.8** | $\uparrow$4.8 |
> > > | Geometric Shapes | 17.6 | **20.0** | $\uparrow$2.4 |
> > > | Logical Deduction (seven objects) | 43.2 | **45.2** | $\uparrow$2.0 |
> > >  | Multi-Step Arithmetic [Two] | 61.6 | **64.0** | $\uparrow$2.4 |
> > > | Tracking Shuffled Objects (seven objects) | 31.6 | **36.0** | $\uparrow$4.4 |
> > >
> > >
> > > All of these experiments were conducted with `gpt-3.5-turbo-0316` (ChatGPT), with a temperature of 0.0, and without any task-specific prompts (all tasks used the same instruction prompts). We believe that task-specific prompts would further increase the gain of Self-Refine compared to the base model.
> > >
> > > We remain open to further feedback and would greatly appreciate any additional insights and questions you might have.

---

> > > > ### Comment · Reviewer_UJCc · 2023-08-17
> > > >
> > > > Thank you.
> > > > I have adjusted my scoring accordingly.

---

> > > > > ### Author Response · Authors · 2023-08-17
> > > > > **Thank you for increasing your score!**
> > > > >
> > > > > Thank you for increasing your score!
> > > > >
> > > > > Please let us know if you have any more questions before the end of the discussion period.

---

### Official Review · Reviewer_mJq7 · 2023-07-05

**Soundness:** 3 good
**Presentation:** 4 excellent
**Contribution:** 3 good
**Rating:** 6
**Confidence:** 4

**Summary:**

This paper proposes a Self-Refine framework for improving initial outputs from LLMs. Given an input, Self-Refine generates feedback and refines its outputs iteratively. Without additional training cost or human effort, Self-Refine outperforms baselines with the same large language model that generated output draft previously. Specifically, Self-Refine can be divided into the following three steps: initial generation, feedback, and refinement. These three steps are accomplished through task-specific in-context learning (i.e., few-shot prompting).

The authors evaluate Self-Refine on seven diverse tasks (including generation and reasoning tasks). The experimental results demonstrate the superiority of the proposed method, which significantly outperforms baselines on automated metrics or GPT-4-pref (high correlation with human preference on a subset). Subsequent analysis shows that LLMs across different model sizes consistently benefit from Self-Refine. Compared with generic feedback, specific and constructive feedback performs better. Moreover, other analytical experiments have proposed valuable conclusions from different perspectives.

**Strengths:**

1. The impressive experimental results show superior performance to the baseline.

2. The authors have done a wealth of analytical experiments, including the impact of the feedback quality and iterations, effectiveness across different scales of models, etc., which are very meaningful.

3. This well-written paper demonstrates Self-Refine implemented by the authors to generate better outputs with the same LLM.

**Weaknesses:**

1. Implementing iterative few-shot prompting-based Self-Refine may require a high computational cost. However, this is a minor issue since the extent of performance improvement indicates that the cost is worthwhile.

2. Self-Refine doesn't work with weaker models, as it requires the models to have good few-shot modeling or instruction-following abilities.

**Questions:**

The GPT series models used in this paper have good instruction-following abilities. I am interested in how this method performs with zero-shot prompting (only instructions), which would facilitate the community in making trade-offs between performance and cost.

**Limitations:**

There is no potential negative societal impact.

---

> ### Author Rebuttal · Authors · 2023-08-10
>
> Thank you for taking the time to review our paper! We were happy to read that you appreciated our main points: that Self-Refine works without additional training cost or human effort, that Self-Refine benefits diverse tasks including generation and reasoning, that Self-Refine can even improve state-of-the-art LLMs at test time, and that experimental results are impressive.
>
> We think that all your questions are addressable within this discussion period. Please see our response below. We will love to address additional questions during the discussion period if anything is unclear.
>
>
> ---
>
> **Implementing iterative few-shot prompting-based Self-Refine may require a high computational cost. However, this is a minor issue since the extent of performance improvement indicates that the cost is worthwhile.**
>
>
>
> We agree that the extent of performance improvement indicates that the cost is worthwhile. Besides the performance improvement, we would like to highlight that:
>
> 1. The first few iterations provide most of the gains, so even 2 or 3 iterations can provide large gains. Figure 4 further analyzes this tradeoff.
> 2. Our approach is applied only at inference, and inference is getting cheaper over time.
>
> In addition, various approaches such as [Chain-of-Thought (Wei et al., NeurIPS’2022)](https://arxiv.org/pdf/2201.11903), [“Least-to-most” (Zhou et al., ICLR’2023)](https://openreview.net/pdf?id=WZH7099tgfM) and [Self-Consistency (ICLR’2023)](https://openreview.net/pdf?id=WZH7099tgfM) provide similar trade-offs: *unlocking emerging capabilities of LLMs, at a slightly higher inference cost*. Finally, we note that Self-Refine presents a trade-off between cost and performance. That is, even two iterations of self-refine offers better results compared to initial responses from these models (see Page 6-Fig 4).
>
>
> ---
>
> **Self-Refine doesn't work with weaker models, as it requires the models to have good few-shot modeling or instruction-following abilities.**
>
> It seems that as reported in other papers such as [Emergent Abilities (Wei et al, 2022)](https://arxiv.org/pdf/2206.07682.pdf), there are abilities that only sufficiently strong/large models exhibit. We believe that Self-Refine is a kind of emergent ability, and in these sufficiently strong models, Self-Refine can significantly further improve their outputs. However, we anticipate that such strong models will be widely available soon. Please see results on the open-access LLAMA-2 model in the global response.
>
> ---
>
> **How does Self-Refine perform with zero-shot prompting (only instructions)?**
>
>
> Few-shot prompting eases the parsing of response, as it shapes the generated content to an easy to extract format. Following the reviewer’s suggestions, we conduct instruction-only experiments where instead of giving few-shot examples, we used instructions at each stage of Self-Refine. The results are shown below for gpt-3.5-turbo-0613, and in the global response for LLaMA-2 70B, and show that Self-Refine continues to be effective in the instruction-only setup
>
> | Experiment             | Base  | Self-Refine (zero-shot) | Equally good |
> |------------------------|-------|-------------|--------------|
> | Acronym Generation| 16.66%| **44.8**%       | 38.5%        |
> | Constrained generation | 41.5% | **46**%         | 12%          |
> | Sentiment Reversal                 | 4.4%  | **71.4**%       | 16.2%        |
> | Math Reasoning (GSM8k)                    | 22.06%| **59**%         | -          |
> | Dialogue Response Generation | 23% | **48.8**% | 22.8% |

---

> > ### Comment · Reviewer_mJq7 · 2023-08-18
> >
> > Thank you for the detailed reply!

---

> > > ### Author Response · Authors · 2023-08-18
> > > **Thanks for your response**
> > >
> > > Thanks for your note, and for considering our response. We hope to have addressed your concerns and questions with the newly reported results.
> > >
> > > Please let us know if you have any more questions before the end of the discussion period.

---

### Official Review · Reviewer_artr · 2023-07-21

**Soundness:** 3 good
**Presentation:** 3 good
**Contribution:** 3 good
**Rating:** 7
**Confidence:** 4

**Summary:**

The paper introduces Self-Refine, an approach for improving Large language models (LLMs) through self feedback and refinement. The method does not require further training, and uses a single LLM as the generator, refiner, and feedback provider. The authors evaluate the method on seven NLG tasks, including dialog response generation and mathematical reasoning, using multiple GPT models, i.e., GPT-3.5, ChatGPT, and GPT-4. Experimental results show that Self-Refine greatly improves model performance in terms of both humans and automatic evaluation.

**I have read authors' rebuttal as well as other reviewers' comments. I have increased my score from 6 to 7.**

**Strengths:**

1. This paper is well motivated and well written.
2. The proposed method is simple and free of training yet shows great improvement over tasks.

**Weaknesses:**

1. The major concern is that the proposed method can be very costly, in particular when the iteration requires multiple steps.
2. The current paradigm seems to highly rely on the great ability of GPTs, and can not work well for other smaller models, such as Vicuna as discussed in the paper.

**Questions:**

NA

**Limitations:**

Yes.

---

> ### Author Rebuttal · Authors · 2023-08-10
>
> Thank you for taking the time to review our paper! We were happy to read that you appreciated our main points: that Self-Refine does not require further training, that Self-Refine benefits a variety of tasks, that Self-Refine can even improve state-of-the-art LLMs at test time, and that Self-Refine greatly improves model performance.
>
> We think that all your questions are addressable within this discussion period. Please see our response below. We will love to address additional questions during the discussion period if anything is unclear.
>
>
> ---
>
> **The major concern is that the proposed method can be very costly, in particular when the iteration requires multiple steps.**
>
>
> While Self-Refine does employ multiple inference iterations, characterizing it as “*very* costly” may not fully capture its nuances. Self-Refine is exclusive to inference—a domain where costs are progressively declining. Various prompting approaches such as [Chain-of-Thought (Wei et al., NeurIPS’2022)](https://arxiv.org/pdf/2201.11903), [“Least-to-most” (Zhou et al., ICLR’2023)](https://openreview.net/pdf?id=WZH7099tgfM) and [Self-Consistency (ICLR’2023)](https://openreview.net/pdf?id=WZH7099tgfM) provide a similar tradeoff: unlocking a model’s full potential and getting better performance, at the cost of longer outputs or querying the model multiple times.
>
> Our approach requires querying the model multiple times during inference, but we believe that in most cases, this is a reasonable price to pay for improving the outputs of the newest and largest models. Notably, Self-Refine substantially improves the quality of responses in LLMs, as noted by [Reviewer mJq7](https://openreview.net/forum?id=S37hOerQLB&noteId=vqWj3bEgMe) and shown in our Table 1.  Further, the tradeoff between the number of iterations (which is proportional to cost) and performance that we analyze in Figure 4 shows that the first few iterations provide most of the gains. So, if one has a limited compute budget, even using a small constant number of iterations provides significant benefits.
>
>
>
> ---
>
>
> **The current paradigm seems to highly rely on the great ability of GPTs, and cannot work well for other smaller models, such as Vicuna, as discussed in the paper.**
>
> Recent papers, such as [Emergent Abilities (Wei et al, 2022)](https://arxiv.org/pdf/2206.07682.pdf), show that there are abilities that only sufficiently strong/large models exhibit. We believe that Self-Refine is a kind of emergent ability, and in these sufficiently strong models, Self-Refine can significantly further improve their outputs. Our experiments with LLAMA2 (please see the global response) demonstrate that even publicly available models now possess these capabilities. As we anticipate the release of even stronger models to the public in the future, the potential and applicability of Self-Refine are set to expand.

---

> > ### Comment · Reviewer_artr · 2023-08-17
> >
> > Thank the authors for their reply.
> > I have increased my score from 6 to 7.

---

> > > ### Author Response · Authors · 2023-08-17
> > > **Thank you for increasing your score!**
> > >
> > > Thank you for increasing your score!
> > >
> > > Please let us know if you have any more questions before the end of the discussion period.

---

### Author Rebuttal · Authors · 2023-08-10

We would like to thank all the reviewers for their valuable feedback. We are encouraged that they find our approach well-motivated (Reviewers [artr](https://openreview.net/forum?id=S37hOerQLB&noteId=QYxVBerAcu), [YNkB](https://openreview.net/forum?id=S37hOerQLB&noteId=lFzaxuj8QC)), efficient in improving LLMs without extra training (Reviewers [artr](https://openreview.net/forum?id=S37hOerQLB&noteId=QYxVBerAcu), [mJq7](https://openreview.net/forum?id=S37hOerQLB&noteId=vqWj3bEgMe), [UJCc](https://openreview.net/forum?id=S37hOerQLB&noteId=9XPd5By9bN)), with comprehensive results surpassing baselines (Reviewers [mJq7](https://openreview.net/forum?id=S37hOerQLB&noteId=vqWj3bEgMe), [YNkB](https://openreview.net/forum?id=S37hOerQLB&noteId=lFzaxuj8QC)), and straightforward and “conceptually neat” (Reviewers [UJCc](https://openreview.net/forum?id=S37hOerQLB&noteId=9XPd5By9bN), [YNkB](https://openreview.net/forum?id=S37hOerQLB&noteId=lFzaxuj8QC)). We have addressed reviewers’ comments individually and look forward to a fruitful interaction during the author response period.


---

**New Results Reported**

We provide all the new results that were requested:
- [Self-Refine in Zero-shot/Instruction-only mode](https://openreview.net/forum?id=S37hOerQLB&noteId=ClZMzlW7nC) (finding: self-refine works well regardless of the prompting approach).
- Results on open-access model LLAMA-2 (finding: Self-Refine provides promising gains even with open-access models)
- [Additional evaluation using Claude-2](https://openreview.net/forum?id=S37hOerQLB&noteId=gk0JpkcMOR) (finding: that self-refine performs well across all metrics).

We also address the other concerns including cost of multiple iterations by discussing the performance cost tradeoff, ensuring blindness in the human evaluation protocol, and address the other clarifications. We will add these clarifications and details in the next version.


---

* Following questions by reviewers `mJq7` and `artr`,
we benchmarked Self-Refine on the open-access model [LLAMA-2](https://ai.meta.com/research/publications/llama-2-open-foundation-and-fine-tuned-chat-model) in an **instruction-only** setting (no few-shot prompts, only instructions). The results show that Self-Refine continues to be effective on an open-access model and without few-shot examples at all. Given these performance metrics, alongside anticipated advancements in hardware, we anticipate the broad and cost-effective applicability of Self-Refine.


Instruction-only (zero-shot) results with LLama-2 as the base model:

| Task                | Base         | Self-Refine  | Equally Good |
|---------------------|--------------|--------------|--------------|
| **Acronyms**        | 22.30      | **53.08**      | 22.30     |
| **Yelp**            | 13.2         | **60.8**         | 26          |
| **Response Generation** | 11.2       | **20.4**          | 54.6         |
| **GSM-8k**          | 37.6         | **37.8 (41 with Oracle)** | N/A   |

---

### Decision · Program_Chairs · 2023-09-21

**Decision:**

Accept (poster)

**Comment:**

This work presents an iterative method to improve LLM performance, where the LLM first produces a response and then produces feedback for itself to improve (before producing another, hopefully better response).

The reviewers liked the approach and comprehensive evaluation as well as the clarity of presentation in general. However, they also raised some concerns: practical cost of the method (monetary and effort), the fact that it may not work with weaker / smaller models, use of GPT4 and humans in evaluations.

The authors provided responses to all reviewers and also provided new results, all of which seemed to address the main concerns the reveries raised. For the above reasons I am happy to accept his paper.